# scRNA-sequencing in chick suggests a probabilistic model for cell fate allocation at the neural plate border

Alexandre P Thiery[1], Ailin Leticia Buzzi[1], Eva Hamrud[1], Chris Cheshire[2], Nicholas M Luscombe[2†], James Briscoe[2], Andrea Streit[1]*

[1]Centre for Craniofacial and Regenerative Biology, Faculty of Dentistry, Oral and Craniofacial Sciences, King's College London, London, United Kingdom; [2]Bioinformatics and Computational Biology Laboratory, The Francis Crick Institute, London, United Kingdom

*For correspondence:
andrea.streit@kcl.ac.uk

Present address: †Genomics and Regulatory Systems Unit, Okinawa Institute of Science and Technology Graduate University, Okinawa, Japan

Competing interest: The authors declare that no competing interests exist.

**Abstract** The vertebrate 'neural plate border' is a transient territory located at the edge of the neural plate containing precursors for all ectodermal derivatives: the neural plate, neural crest, placodes and epidermis. Elegant functional experiments in a range of vertebrate models have provided an in-depth understanding of gene regulatory interactions within the ectoderm. However, these experiments conducted at tissue level raise seemingly contradictory models for fate allocation of individual cells. Here, we carry out single cell RNA sequencing of chick ectoderm from primitive streak to neurulation stage, to explore cell state diversity and heterogeneity. We characterise the dynamics of gene modules, allowing us to model the order of molecular events which take place as ectodermal fates segregate. Furthermore, we find that genes previously classified as neural plate border 'specifiers' typically exhibit dynamic expression patterns and are enriched in either neural, neural crest or placodal fates, revealing that the neural plate border should be seen as a heterogeneous ectodermal territory and not a discrete transitional transcriptional state. Analysis of neural, neural crest and placodal markers reveals that individual NPB cells co-express competing transcriptional programmes suggesting that their ultimate identify is not yet fixed. This population of 'border located undecided progenitors' (BLUPs) gradually diminishes as cell fate decisions take place. Considering our findings, we propose a probabilistic model for cell fate choice at the neural plate border. Our data suggest that the probability of a progenitor's daughters to contribute to a given ectodermal derivative is related to the balance of competing transcriptional programmes, which in turn are regulated by the spatiotemporal position of a progenitor.

## Editor's evaluation

This is a useful study that describes the heterogeneous expression of a very large number of genes ostensibly associated with the earliest cell fate decisions in the ectoderm of chicken embryos. The evidence is solid, and the authors mostly succeed in presenting their complex strategy of single cell RNA-seq and subsequent data analysis in a compelling way. However, some of the conclusions about the ultimate fates of the differentiating cells remain speculative and any further validation would require functional studies and experimental tests of hypotheses. Overall, the methods applied, and the data generated by this study will be valuable to developmental biologists, especially those who study patterning at the neural plate border in vertebrates.

## Introduction

The vertebrate nervous system is arguably the most complex organ of our body. During development, it arises from only three progenitor populations: the neural plate generates the entire central nervous system, while neural crest and sensory placodes jointly form the peripheral nervous system. Placode derivatives are confined to the head contributing to sense organs and cranial ganglia, while neural crest cells form along the entire body axis to generate various cell types including neurons and glial as well as parts of the craniofacial skeleton, cartilage, and teeth (for review: *Baker and Bronner-Fraser, 1997a*; *Baker and Bronner-Fraser, 2001*; *Grocott et al., 2012*; *Patthey et al., 2014*; *Pla and Monsoro-Burq, 2018*; *Prasad et al., 2019*; *Simões-Costa and Bronner, 2015*).

The neural plate becomes morphologically distinct shortly after gastrulation. At its edge, the co-expression of both neural and non-neural markers broadly demarcate the neural plate border (NPB): an ectodermal territory which has been identified in a range of vertebrate species (*Bhat et al., 2013*; *Kwon et al., 2010*; *Pla and Monsoro-Burq, 2018*; *Roellig et al., 2017*; *Streit, 2002*; *Thiery et al., 2020*). Lineage tracing has shown that the NPB contains a mixture of neural, neural crest, placodal, and epidermal precursors (*Bhattacharyya et al., 2004*; *Bronner-Fraser and Fraser, 1988*; *Dutta et al., 2005*; *Kozlowski et al., 1997*; *Pieper et al., 2011*; *Streit, 2002*; *Xu et al., 2008*). As the neural plate invaginates, placode precursors remain in the surface ectoderm, while neural crest progenitors are largely incorporated into neural folds from where they delaminate and migrate extensively. Although the induction of these three fates has been studied for decades, some of the most fundamental questions remain unresolved. For example, we do not know when and how these lineages segregate, whether they do so in a specific order and how individual cells make these decisions.

Modulation of different signalling pathways plays a crucial role in controlling the fate of all ectodermal cells. FGF signalling is required for the specification of neural, neural crest and placode precursors and for positioning the NPB (*Ahrens and Schlosser, 2005*; *Litsiou et al., 2005*; *Londin et al., 2005*; *Monsoro-Burq et al., 2003*; *Streit et al., 2000*; *Streit and Stern, 1999*; *Stuhlmiller and García-Castro, 2012*; *Wilson et al., 2000*; *Yardley and García-Castro, 2012*). While BMP signalling must be inhibited for neural development (*Hemmati-Brivanlou et al., 1994*; *Linker and Stern, 2004*; *Londin et al., 2005*; *Sasai et al., 1995*), it is necessary for placode precursor formation at gastrulation, but must be switched off thereafter (*Ahrens and Schlosser, 2005*; *Kwon et al., 2010*; *Litsiou et al., 2005*). Likewise, neural crest cells need BMP activity early, but unlike placodes, continue to do so as they are firmly established (*Barth et al., 1999*; *Kwon et al., 2010*; *Liem et al., 1995*; *Marchant et al., 1998*; *Steventon et al., 2009*). Finally, Wnt activity is required for neural crest cell induction (*García-Castro et al., 2002*; *Steventon et al., 2009*) but must be inhibited for neural and placode precursors to form (*Heeg-Truesdell and LaBonne, 2006*; *Litsiou et al., 2005*; *Patthey et al., 2008*). These signals activate different transcription factor networks in a temporally and spatially controlled manner, which in turn are thought to impart cell identity and have therefore been termed fate 'specifiers'. For example, *Msx1* and *Pax3* have been implicated as NPB specifiers (*Milet et al., 2013*; *Monsoro-Burq, 2015*; *Plouhinec et al., 2014*), whereas *Pax7* (*Basch et al., 2006*) and *Six1* (*Brugmann et al., 2004*; *Chen et al., 2009*; *Christophorou et al., 2009*; *Ozaki et al., 2004*) control neural crest and placode precursor formation, respectively. These factors appear to act in feed forward loops, upregulating their own expression while simultaneously repressing factors that specify alternative fates (for review: *Grocott et al., 2012*; *Thiery et al., 2020*).

Based on these studies combined with tissue transplantation experiments, various models for the segregation of neural, neural crest, and placodal fates have emerged (for review: *Schlosser, 2006*; *Schlosser, 2014*; *Thiery et al., 2020*), some proposing that the NPB is an area of 'indecision'. Most recently, two apparently contradictory models have been discussed more widely. The 'binary competence model' proposes that while a territory containing neural crest and placodal precursors exists at the time of gastrulation, this territory is rapidly subdivided into neural/neural crest competent and placode/epidermal competent ectoderm through changes in the regulatory relationships of the transcription factors expressed (sequence of cell fate decisions: [epidermal-placodal]-[NC-neural]) (*Ahrens and Schlosser, 2005*; *Pieper et al., 2012*; *Schlosser, 2014*; *Maharana and Schlosser, 2018*). This implies a closer relationship of neural crest cells with the central nervous system and of placodes with the epidermis. In contrast, the 'NPB model' suggests that cells at the border of the neural plate have mixed identity and retain the ability to generate all ectodermal derivatives until after neurulation begins (*Baker and Bronner-Fraser, 1997b*; *Roellig et al., 2017*; *Streit and Stern, 1999*). Other

studies in *Xenopus* have taken this idea a step further, revealing that unlike other ectodermal cells, cells at the NPB retain pluripotency markers which is key for their multipotency (*Buitrago-Delgado et al., 2015*; *Prasad et al., 2012*). In support of this idea, the NPB region is initially characterised by co-expression of neural and non-neural markers (for review: *Grocott et al., 2012*; *Pla and Monsoro-Burq, 2018*; *Thiery et al., 2020*).

The distinguishing feature of the binary competence and NPB models is the proposed order in which cell fate decisions are made. The NPB model does not imply a specific sequence of decision events, whilst the binary competence model proposes that NPB progenitors transition through an intermediate fate restricted state. While both models are attractive and provide working hypotheses, the underlying experiments were analysed at cell population, rather than at single cell level. Furthermore, the literature is heavily biased towards investigating formation of neural crest cells, which are absent in the most anterior NPB. Therefore, these models do not consider the differences in potential fate decisions along the antero-posterior axis but propose that all cells at the neural plate border face the same set of discrete choices. Recent findings have proposed that prior to fate commitment, cells show increasing levels of heterogeneity in gene expression and do not exhibit a definitive transcriptional signature (*Soldatov et al., 2019*; *Subkhankulova et al., 2023*). It is therefore possible that cells at the NPB include such undecided progenitor cells.

Studies in chick have begun to characterise cell heterogeneity at the NPB at a single-cell level revealing that factors previously thought to be fate-specific are co-expressed in individual cells even at late neural tube stages (*Roellig et al., 2017*; *Williams et al., 2022*). Furthermore, classical lineage tracing reveals that individual ectodermal cells can give rise to both neural and neural crest even at fairly late stages (*Bronner-Fraser and Fraser, 1988*), while more a more recent study using a neural-specific *Sox2* enhancer shows that Sox2+ cells can contribute not only to neural tissue but also to neural crest and epidermis (*Roellig et al., 2017*). However, these studies only surveyed a relatively small number of ectodermal cells and, focusing on neural crest cells, did not include data for definitive placodal cells. It is therefore challenging to assess how these fates segregate. Nevertheless, these findings suggest that cell fate allocation at the NPB occurs over a much longer developmental period than previously thought and that some NPB cells retain plasticity until early neurulation.

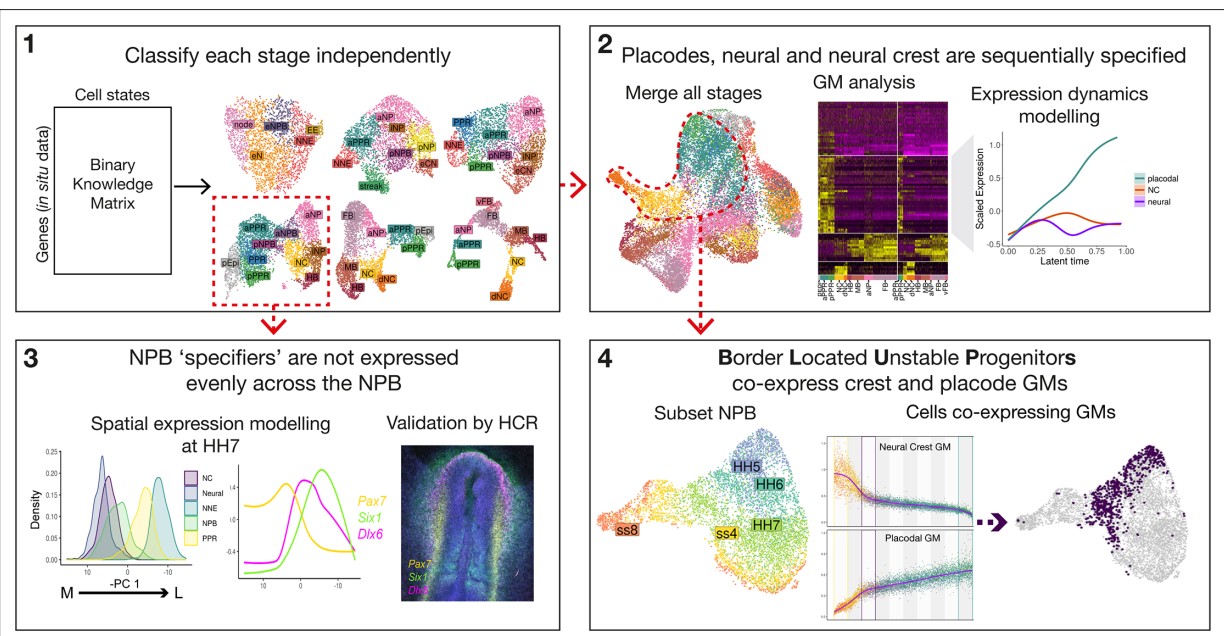

**Figure 1.** Workflow and summary of findings. Workflow schematic highlighting key findings from our analysis. Dotted red lines and arrows illustrate data moving from one part of the analysis to the next. The main analysis can be broken into 4 main steps: (1) classify cell states at each developmental stage independently using a binary knowledge matrix; (2) merge cells from all stages to identify global gene modules characterising ectodermal lineage segregation; (3) subset cells at HH7 to model spatial expression patterns across the mediolateral axis; and (4) subset cells at the NPB and identify 'border located unstable progenitors' through co-expression of placodal and neural crest gene modules.

Here, we explore the transcriptional changes as cells transit from the neural plate border at early primitive streak to definitive neural, neural crest and placodal fates at single cell level (for analysis workflow see *Figure 1*). In total, we present single-cell RNAseq (scRNAseq) data for 17,992 high-quality cells. Rather than using 2–3 molecular markers to assign cell identity, we established a binary knowledge matrix using previously published in situ hybridisation expression data. This allows us to classify cells at each developmental stage, despite the high levels of heterogeneity observed. We find that prior to head fold stages cells do not fall into transcriptional distinct groups but are characterised by graded and partially overlapping gene expression. An early cell population located at the NPB (eNPB) can be identified by co-expression of neural and non-neural genes prior to the upregulation of fate specifiers. As development proceeds, ectodermal cell diversity increases, with a division of neural and non-neural fates becoming apparent at the first somite stage together with the emergence of neural crest cells. By the 8-somite stage, neural, neural crest and placodal cells are largely segregated, although even at this late stage we find cells with heterogenous expression profiles.

Using a gene module approach, we identify dynamic changes in groups of genes that are characteristic for neural, neural crest and placodal fates. Comparing co-expression of whole gene modules (rather than individual transcripts) reveals that many cells at the NPB co-express gene modules which later become restricted to placodal or neural crest lineages. These cells may have the potential to give rise to both (or more) fates and we therefore refer to them as 'Border-Located Undecided Progenitors' (BLUPs). Interestingly, as progenitors segregate into their respective fates, we no longer identify NPB clusters but continue to find BLUPs. BLUPs connect segregating cell populations, resembling previously characterised bridge cells in the differentiating murine neural crest which exhibit high heterogeneity prior to segregation (*Soldatov et al., 2019*). Here, we propose that BLUPs continue to be plastic until neural tube closure, maintaining the ability to contribute to different fates. Overall, our analysis highlights that although the NPB demarcates an anatomical region, it is not a discrete transcriptional state. Instead, we identify BLUPs which we predict to be in an undecided state and to retain the potential to contribute toward alternative ectodermal fates.

## Results
### Classification of ectodermal cells using a binary knowledge matrix of known marker genes

Neural, neural crest and placode cells arise from the embryonic ectoderm. However, how individual progenitors acquire their terminal fates and the sequence and timing of their segregation remain to be elucidated. To capture the molecular changes during this process, we carried out 10 x Genomics single-cell mRNA sequencing (scRNAseq) on cells taken from a broad region of the chick epiblast at six developmental stages: HH4⁻/4 (primitive streak), HH5 (head process), HH6 (head fold), HH7 (1 somite), somite stage (ss) 4 and ss8 (*Figure 2A*). These stages were chosen because they encompass the initial formation of the neural plate to the beginning of neurulation when these fate decisions are thought to take place. After integration across batches, stringent quality control and removal of contaminating cell populations (see Materials and methods and *Figure 2—figure supplement 1A-I* ), we captured a total of 17,992 high-quality cells with approximately 2500–4000 cells per stage and an average of 3924 genes per cell. Dimensionality reduction and UMAP embedding reveals that cells at later stages displayed higher cell diversity compared with earlier stages where most cells were transcriptionally similar (*Figure 2B–E* and *Figure 3A–D*).

Typically, scRNAseq cell states are classified using a small number of definitive marker genes which exhibit mutually exclusive expression patterns. However, this approach is difficult to apply to the NPB for several reasons. Firstly, the early NPB cannot be classified using definitive markers as it emerges in the region of overlap between neural and non-neural genes. Secondly, cells at the NPB are thought to be in a transitory state before they acquire their definitive ectodermal fate (neural, neural crest, placodal and epidermal). Finally, cells at the NPB appear to be highly heterogeneous, with precursors for the different fates found intermingled, even at late stages of neurulation (*Bhattacharyya et al., 2004*; *Bronner-Fraser and Fraser, 1988*; *Ezin et al., 2009*; *Roellig et al., 2017*; *Streit, 2002*; *Williams et al., 2022*; *Xu et al., 2008*). To overcome these challenges, we categorised cells using a binary knowledge matrix based on expression data obtained from the literature (*Supplementary file 1*). We examined available in situ hybridisation data of 76 genes with known regionalised expression

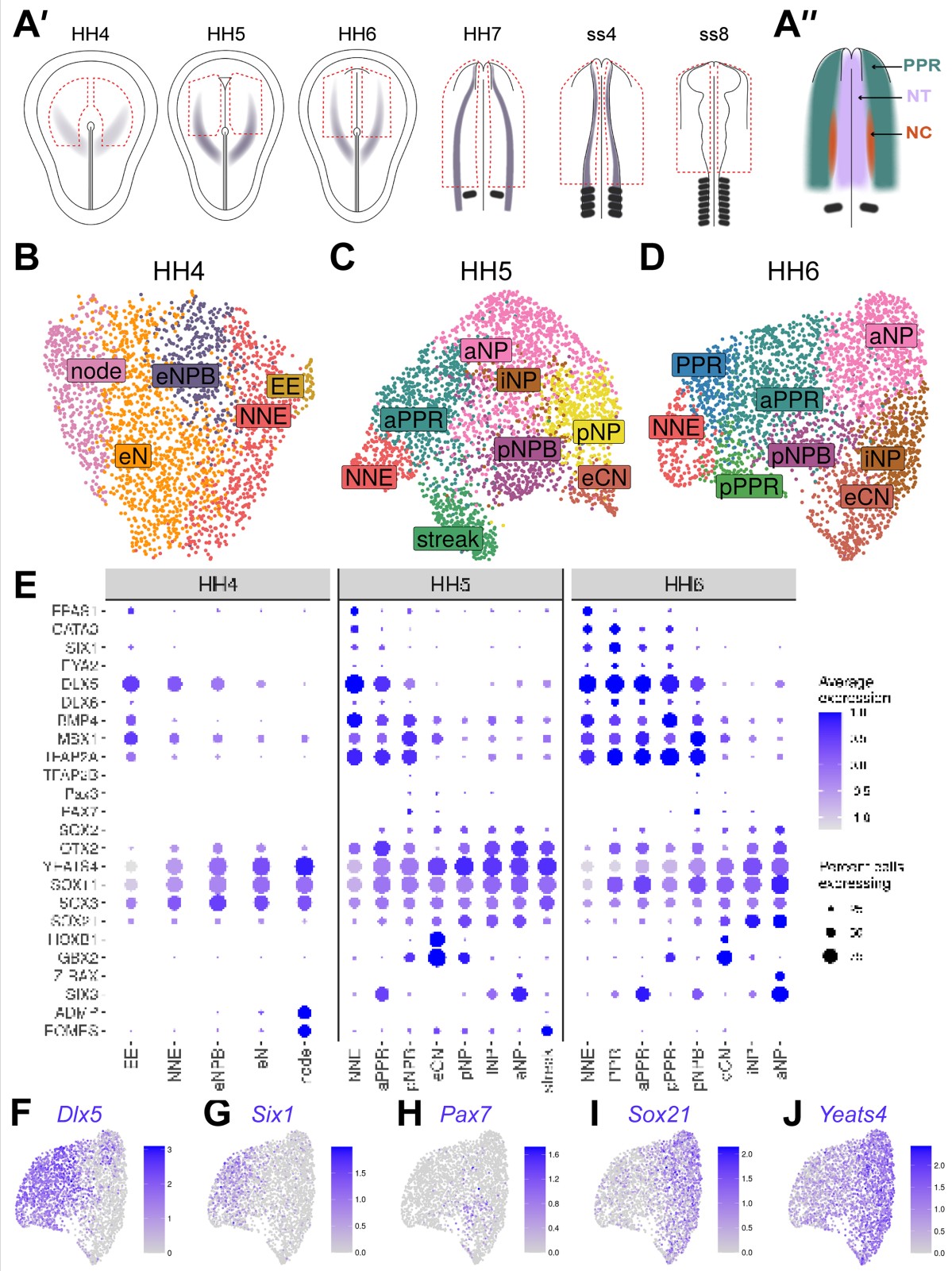

**Figure 2.** Cells at stages HH4 to HH6 reflect the anterior-posterior and medio-lateral axes in the embryo. (**A'**) Dorsal view schematics of chick embryo at stages HH4-ss8 depicting the ectodermal tissue region dissected for 10 x scRNAseq (red dotted line). Purple shading illustrates previously characterized region of *Pax7* expression (***Basch et al., 2006***). (**A''**) Schematic of a 1-somite stage (HH7) chick embryo illustrating the pre-placodal region (PPR), neural crest (neural crest) and neural tube (NT). (**B–D**) UMAP plots for cells collected at HH4⁻/4 (primitive streak), HH5 (head process), HH6 (head fold) stages.

*Figure 2 continued on next page*

*Figure 2 continued*

Cell clusters are coloured and labelled based on their semi-unbiased cell state classifications (binary knowledge matrix available in *Supplementary file 1*). (**E**) Dot plots displaying the average expression of key marker genes across cell states at HH4, HH5 and HH6. The size of the dots represents the number of cells expressing a given gene within each cell population. (**F**) Feature plots revealing the mediolateral expression of key marker genes at HH6 and their overlapping expression.

The online version of this article includes the following figure supplement(s) for figure 2:

**Figure supplement 1.** Data analysis and quality control.

**Figure supplement 2.** Feature plots showing expression of marker genes on UMAPs of developmental stages HH4 (**A**), HH5 (**B**), and HH6 (**C**).

patterns and binarized their expression within each defined cell state. This allowed us to define a total of 24 cell states spanning primitive streak to neural tube stages. Using this approach, we classify NPB cells not only by the expression of previously characterised NPB specifiers *Pax7* and *Tfap2a*, but also by the overlapping expression of early neural and non-neural genes.

We first unbiasedly clustered cells from each developmental stage at a high resolution to ensure capturing the full diversity of cell states (*Figure 2—figure supplement 1J*). Clusters were then classified based on the similarity between their gene expression and the knowledge matrix (see Materials and methods). This classification approach successfully identified 22 distinct cell states within the ectoderm across all developmental stages. Early NPB and early neural cell populations can already be identified at primitive streak stages, while placode progenitor and neural crest states emerge at head process and ss4, respectively (*Figure 2B* and *Figure 3A–B*).

## Cell states emerging from primitive streak to head fold stages

To explore the temporal changes in cell diversity within the embryonic ectoderm, we compared cell states present at different stages and identified the time point when each emerged. We first examined stages HH4-HH6 to characterise molecular events as neural, placodal and neural crest transcriptional signatures are thought to be established.

### An early NPB population is identified at primitive streak stages

By primitive streak stage, mediolateral (M-L) gradients of molecular markers demarcate pre-neural and non-neural territories within the chick epiblast (*Pera et al., 1999*; *Rex et al., 1997*; *Sheng and Stern, 1999*; *Streit et al., 2000*; *Streit et al., 1998*). At HH4, unbiased clustering clearly organises cells in a pattern reminiscent of this M-L axis (*Figure 2B*; from left to right) as evidenced by the expression of node (*ADMP*), pre-neural (*Yeats4, Sox3, Sox11*), and non-neural (*Dlx5/6, Bmp4, Msx1, Tfap2a*) genes (*Figure 2E* and *Figure 2—figure supplement 2* ). Cell states can be broadly characterised based on combinatorial gene expression (*Figure 2*) and we identify five cell states: node, early neural (eN), early neural plate border (eNPB), non-neural ectoderm (NNE) and extraembryonic ectoderm (EE) (*Figure 2B*). While some genes are shared between many cells, early neural plate cells are defined by high levels of pre-neural transcripts while the non-neural and extraembryonic ectoderm are characterised by increased non-neural gene expression. Notably, we identify an early NPB population which co-expresses both neural and non-neural genes. Thus, early NPB cells are already apparent prior to the differential upregulation of genes previously characterised as NPB specifiers (*Msx1, Pax7, Tfap2a, Dlx5/6, Pax3*) (*Hong et al., 2007*; *Maharana and Schlosser, 2018*; *Monsoro-Burq et al., 2005*; *Pla and Monsoro-Burq, 2018*) (for review: *Sauka-Spengler and Bronner-Fraser, 2008*; *Simões-Costa and Bronner, 2015*; *Thiery et al., 2020*).

### The first emerging NPB and PPR cells have regional identity

At HH5, cell positions in the UMAP continue to reflect organisation along the M-L axis in the embryo (*Figure 2C*), but new transcriptional states emerge. Whilst non-neural ectoderm cells display a similar expression profile at HH5 compared to HH4, neural plate cells differ from early neural cells in their expression of definitive neural (e.g. *Sox2*) and regional markers (*Figure 2E*; *Figure 2—figure supplement 2B*). Like at HH4, many cells continue to co-express neural and non-neural transcripts (*Figure 2—figure supplement 2B*); however, at HH5 we observe the upregulation of PPR genes (*Six1, Eya2*; *Figure 2C and E*: aPPR, anterior PPR) and for the first time so-called NPB specifiers (*Msx1, Pax3, Pax7*; *Figure 2C and E*: pNPB, posterior NPB). These observations correlate well with gene

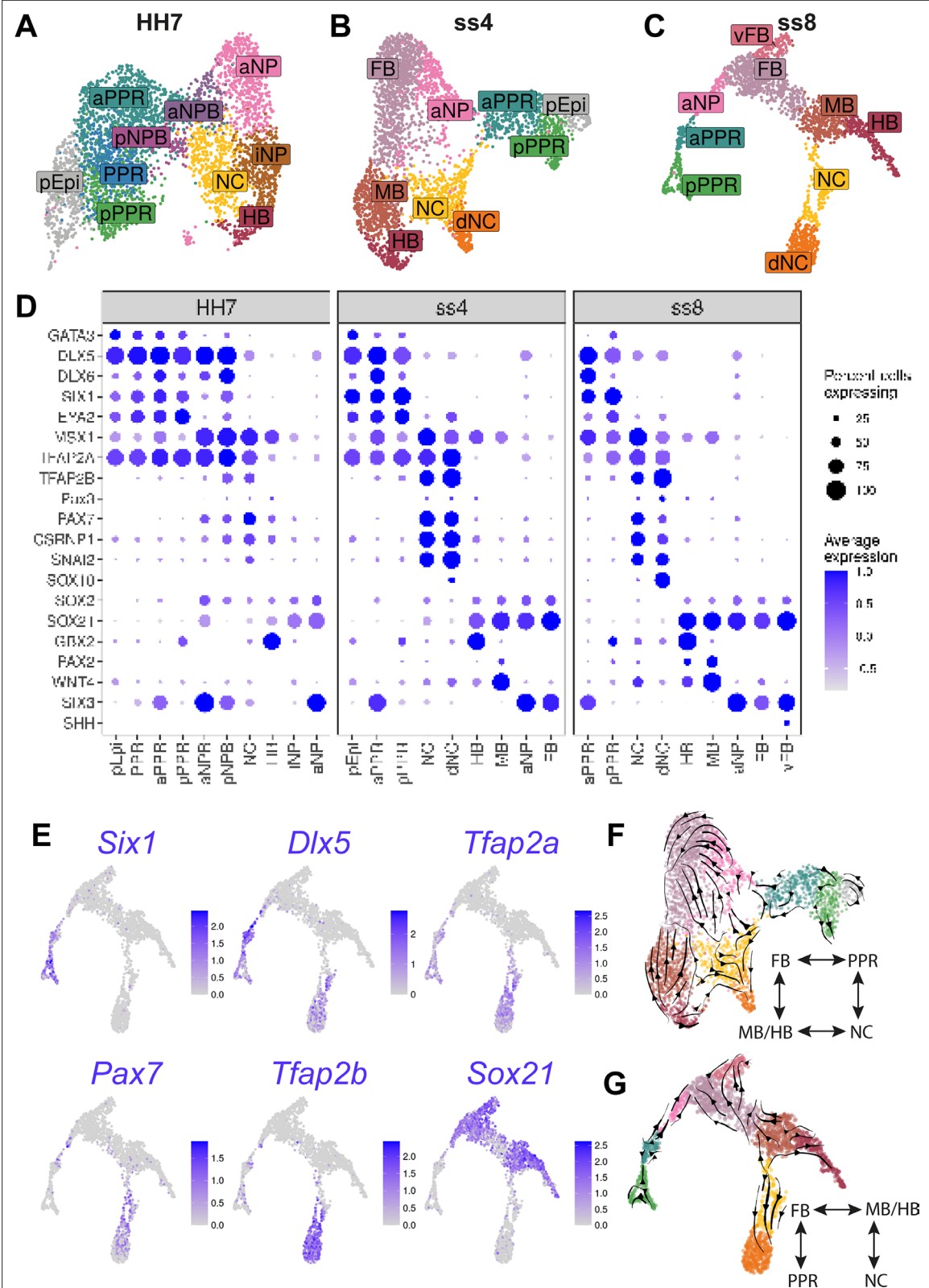

**Figure 3.** Increased cell diversity and lineage segregation from HH7 to ss8. (**A–C**) UMAP plots for cells collected at HH7 (1 somite stage), ss4 (4 somite stage), and ss8 (8 somite stage). Cell clusters are coloured and labelled based on their semi-unbiased cell state classifications (binary knowledge matrix available in ***Supplementary file 1***). (**D**) Dot plots displaying the average expression of key marker genes across cell states at HH7, ss4, and ss8. The size of the dots represents the number of cells expressing a given gene within each cell population. (**E**) Feature plots revealing the expression of the

*Figure 3 continued on next page*

*Figure 3 continued*

pioneer factor *Tfap2a* and lineage restricted expression of neural (*Sox21*), placodal (*Six1, Dlx5,*), and neural crest (*Pax7, Tfap2b*) markers. (**F–G**) UMAP plots for ss4 and ss8 overlaid with RNA velocity vectors depicting the predicted directionality of transcriptional change. Cells are coloured by cell state classification (shown in B and C). Schematics summarise the key fate segregation events taking place at these stages predicted by RNA velocity analysis.

The online version of this article includes the following figure supplement(s) for figure 3:

**Figure supplement 1.** Feature plots showing expression of marker genes on UMAPs of developmental stages HH7 (**A**), ss4 (**B**), and ss8 (**C**).

expression in the embryo. *Msx1* is initially expressed in the extraembryonic and non-neural ectoderm, and together with *Pax7* begins to be upregulated at the NPB thereafter (**Basch et al., 2006**; **Streit and Stern, 1999**), while PPR markers are first expressed at HH5. Notably, the NPB initially has a posterior character, while placode precursors have anterior identity based on their expression of *Gbx2* and *Six3*, respectively (**Figure 2E**; see also below).

## Anteroposterior organisation of cells at head process and head fold stages

Cells with distinct anteroposterior (A-P) character are first observed at HH5 through the restricted expression of various homeobox transcription factors. *Rax* and *HoxB1* are highly confined to distinct cell groups, whilst in contrast *Six3* and *Gbx2* exhibit broader negatively correlated expression gradients across cell clusters with some degree of overlap (**Figure 2E** and **Figure 2—figure supplement 2B**). The organisation of cells within the UMAP reflects the A-P expression gradients of these genes in the embryo (**Chapman et al., 2002**; **Hidalgo-Sánchez et al., 2005**; **Ohuchi et al., 1999**; **Paxton et al., 2010**). The overlap of A-P factors was used to distinguish different NP, PPR, and NPB regions during unbiased cell state classification (**Supplementary file 1**).

Within the NP, we can distinguish intermediate neural plate (iNP) states from the anterior and posterior neural plate (aNP, pNP) and from early caudal neural (eCN) states. However, within the PPR and NPB we identify only anterior (aPPR; *Six3*$^+$) and posterior (pNPB; *Gbx2*+) cell states, respectively. While in UMAPs, aPPR cells are located between the NNE and anterior NP, the pNBP cluster resides between the iNP and pPPR clusters (**Figure 2C and E Figure 2—figure supplement 2C**), reflective of both its A-P and M-L position in vivo.

Between HH5 and HH6, posterior *Gbx2*$^+$ PPR (pPPR) cell states emerge (**Figure 2D**), as do PPR cells that appear to lack regional identity. At this stage, *Otx2* and *Gbx2* expression domains in the embryo abut (**Steventon et al., 2012**), suggesting that cells with generic PPR character might be intermingled with those already biased towards axial identity and either retain the potential to contribute to multiple axial levels or will later become intermediate PPR cells.

Together, our data show that the emergence of cells with divergent transcriptional profiles resembles patterning of the ectoderm in vivo. However, unbiased cell clustering forces cells into discrete clusters which may not fully reflect heterogeneity in gene expression and cell diversity. Indeed, our single-cell data reveal the heterogenous nature of gene expression: markers are broadly expressed and are not confined by cluster boundaries or by neighbouring expression domains (**Figure 2F–J**, **Figure 2—figure supplement 2C**). Despite this heterogeneity, our cell classification approach successfully identifies increased ectodermal cell diversity over time, with neural, posterior NPB and anterior PPR cells emerging when the definitive neural plate forms. Importantly, the relative positions of these cell states in the UMAPs reflect their location in the embryo.

## Increasing cell diversity from 1-somite to 8-somite stages

At HH7, the neural folds begin to elevate indicating the start of neurulation, while during early somitogenesis neural crest cells start to migrate and placode precursors begin to diversify. We therefore sought to characterise the transcriptional changes at single cell level as neural, neural crest and placodal fates emerge.

Following clustering, we identified two major superclusters at HH7, one containing neural tube and neural crest and the other placodal and future epidermal clusters (**Figure 3A**). These superclusters appear to be largely separate, both by their UMAP embedding and the restricted expression of definitive fate markers (**Figure 3A and D**). Whereas at earlier stages many cells co-express non-neural and neural markers (**Figure 2E, F and J**), at HH7 cells expressing non-neural/placodal (*Dlx5/6, Six1*) and neural transcripts (*Sox21, Sox2*) are largely confined to their respective supercluster (**Figure 3—figure**

*supplement 1* ). It is at this stage that anterior NPB cells can first be identified transcriptionally, and together with posterior NPB cells they connect the two superclusters (*Figure 3*; aNPB, pNPB).

## The neural crest state emerges at early neurulation

Adjacent to the NPB clusters, we observe a neural crest cell cluster expressing early definitive neural crest markers including *Tfap2b*, *Snai2*, and *Axud1* (*Csrnp1*) alongside NPB markers (*Msx1*, *Pax7*, and *Tfap2a*; *Figure 3D*; NC and *Figure 3—figure supplement 1A*). Thus, neural crest cells first emerge as a distinct cell population at HH7. Although many genes are shared between the NPB and neural crest cell clusters, NPB cells also express PPR and non-neural markers (*Six1*, *Eya2*, and *Dlx5/6*) which are in turn downregulated in neural crest cells (*Figure 3D* and *Figure 3—figure supplement 1A*). Furthermore, we find that *Msx1* and *Pax7* are not strongly expressed within placodal cell clusters but instead are enriched in neural crest progenitors. This observation contests their previously characterised role as pan-NPB specifiers.

## Neural, placodal, and neural crest lineages are largely segregated by ss4

At ss4 and ss8, a NPB cell cluster can no longer be identified using the above criteria. Instead, we observe an increasing diversity of cells classified as placodal, neural crest or neural (*Figure 3B, C and E*). Within the neural cell population, forebrain, midbrain, and hindbrain (FB, MB, HB) clusters are clearly distinguishable. *Six3* and *Gbx2* expression define cells with fore- and hindbrain identity, respectively, whilst the upregulation of the dorsal midbrain marker *Wnt4* (*Hollyday et al., 1995*) and mid/hindbrain boundary marker *Pax2* (*Hidalgo-Sánchez et al., 1999*) highlights midbrain-like cells (*Figure 3D* and *Figure 3—figure supplement 1B,C*). As development progresses, neural, neural crest and placode cells become increasingly distinct, with neural crest cells beginning to express a suite of definitive neural crest markers, including *Sox10* in the delaminating neural crest (dNC) cluster (*Figure 3D*).

Despite the apparent segregation of neural, neural crest and placodal cell states at ss4 and ss8, small cell populations connect PPR, neural crest, forebrain and midbrain clusters, in a four-way pattern reflecting their A-P and M-L segregation in vivo (*Figure 3B and C*). To predict the directionality of transcriptional change within these populations, and hence their potential developmental trajectory, we performed RNA velocity analysis (*Bergen et al., 2020*; *La Manno et al., 2018*). This analysis predicts that they move away from cluster boundaries, towards definitive cell states (*Figure 3F and G*).

Together our data show that whereas cell states at HH4-HH6 exhibit broad overlapping gene expression, fate restricted markers are upregulated from HH5 onwards. At HH7 and ss4, placodal, neural crest and neural states begin to segregate. However, even as the neural tube closes, cells continue to straddle cluster boundaries; these cells may be transcriptionally undecided and therefore still actively undergoing fate decision processes.

## Dynamic changes in gene module expression over time

Many transcriptional regulators and their interactions have been implicated in cell fate decisions as the ectoderm differentiates into its derivatives. However, studies have largely taken a gene candidate approach. To explore the full transcriptional dynamics of both known and new transcriptional regulators during development of neural, neural crest and placodal cells, we modelled the dynamics of whole gene modules rather than individual candidate genes (to visualise the dynamics of individual genes see our ShinyApp: https://shiny.crick.ac.uk/thiery_neural_plate_border/). This allows us to investigate transcriptional dynamics in an unbiased manner whilst simultaneously minimising the loss of potentially important factors.

## Defining neural, neural crest, and placodal gene modules

To investigate the segregation of neural, neural crest and placode fates, we grouped together cells from all developmental stages (*Figure 4A and B*) and sought to identify gene modules that characterise each cell lineage. Gene modules are groups of genes that display similar expression profiles across a dataset. We calculated gene modules using the Antler R package (*Delile et al., 2019*), which iteratively clusters genes hierarchically based on their gene-gene Spearman correlation. This approach reveals 40 modules each showing highly correlated gene expression across all stages (HH4-ss8). To focus on gene modules which may include candidates for fate segregation, we included only those

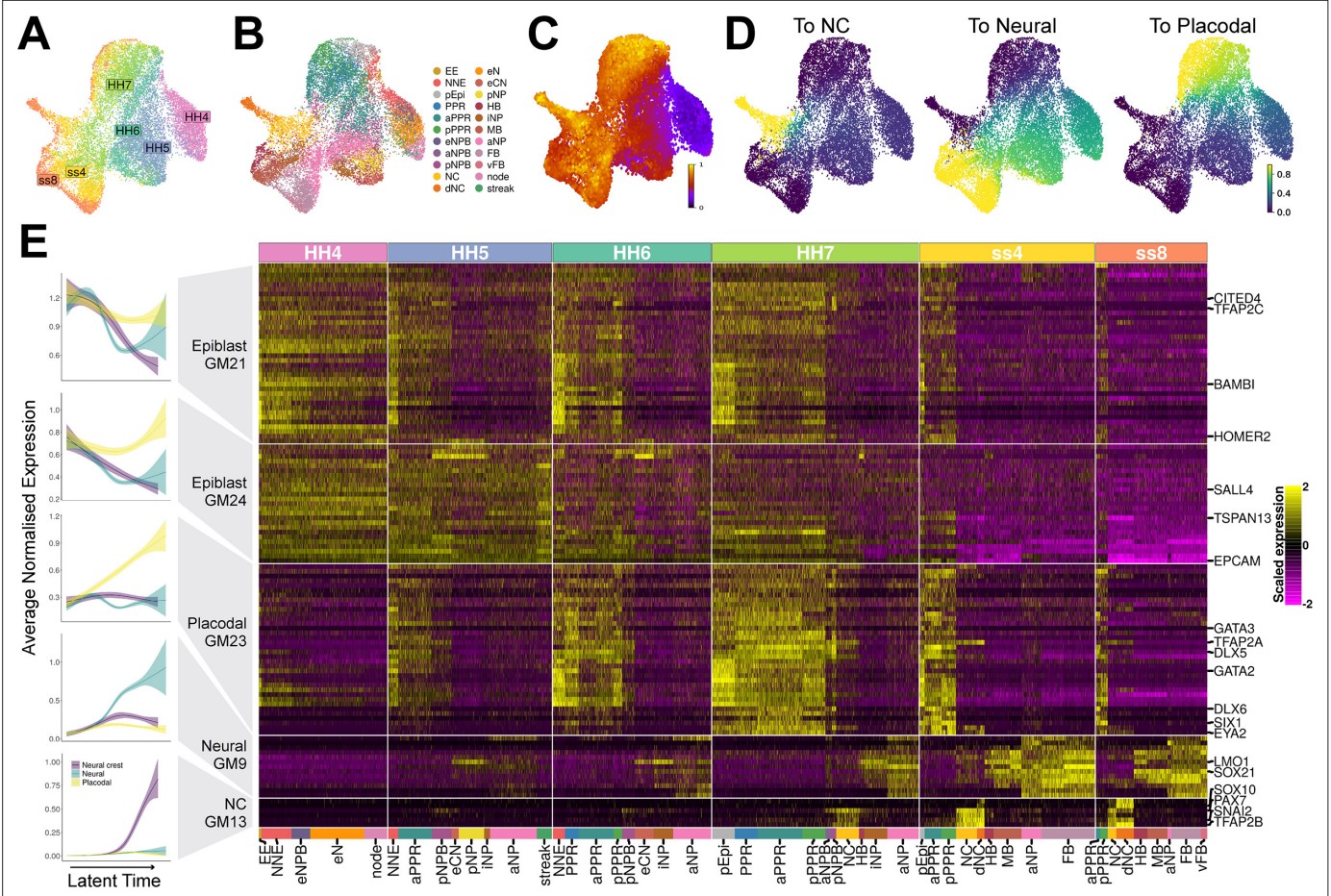

**Figure 4.** Gene module dynamics reveal key differences between the segregation of the PPR, neural crest and neural fates. (**A**) UMAP plot for the full dataset (HH4-ss8 combined) coloured and labelled by stage. (**B**) UMAP plot for the full dataset coloured by cell state classifications. Cell state classifications were calculated independently for each stage and transferred across for visualisation of the full dataset. (**C**) UMAP plot of the full dataset (HH4-ss8) showing cell latent time values. (**D**) UMAP plots of the full dataset showing the fate absorption probabilities of each cell towards one of the three defined terminal states: neural crest, neural and placodal. (**E**) Left: Gene module dynamics plots displaying generalised additive models (GAMs) of average normalised gene module expression across latent time. GAMs are weighted by the fate absorption probability of each cell towards one of the three terminal states (placodal, neural crest and neural). Shaded area represents upper and lower 95% confidence intervals. Right: Heatmap of gene modules that display fate-specific expression (full list of gene modules available in *Figure 4—source data 1*). Gene modules were first unbiasedly filtered based on their differential expression between cell states (see methods and *Figure 4—figure supplement 1A*). Gene modules were further manually filtered based on expression patterns described in the literature (see results section: *Defining neural, neural crest, and placodal gene modules*). Key genes of interest are highlighted on the right (for full gene list see *Figure 4—source data 1*).

The online version of this article includes the following source data and figure supplement(s) for figure 4:

**Source data 1.** Gene lists from gene modules calculated on the full dataset and filtered to include only those differentially expressed between cell states and not sequencing batches, and then further filtered to include only those that are differentially expressed between the neural, neural crest and placodal fates at ss8.

**Figure supplement 1.** Gene modules and pseudo-time analysis.

that are differentially expressed between any cell state and the rest of the dataset, and then filtered to remove those with differential expression between sequencing batches. Finally, since our data show that the segregation of neural, neural crest and placodal cell states is almost complete at ss8 (*Figure 3C*), we kept those gene modules that show differential expression between at least one of the three fates at ss8 (see Materials and methods). This resulted in nine modules that were considered further ( *Figure 4—figure supplement 1* and *Figure 4—source data 1*). Of these, four gene modules (GM21-24) are specifically expressed in the PPR at ss8, one (GM13) in neural crest cells, and

the remaining four (GM7, 9, 11, 20) are upregulated primarily in neural clusters (*Figure 4—figure supplement 1A*).

When examining the genes within each module, we noticed that GM20 and GM22 include genes with widespread expression or expression in the lateral mesoderm and/or the extraembryonic region (*Epas1, Nkx2-5, Tnnc1, Pgk1*; *Adams et al., 2008*; *Bell et al., 2004*; *Liberatore et al., 2000*; *Ota et al., 2007*; *Figure 4—source data 1*). We therefore removed these modules from subsequent analysis. GM7 contains well characterised anterior markers including *Otx2, Six3*, and *Pax6*, whilst GM11 contains midbrain/hindbrain markers including *Pax2, Nkx6-2,* and *Wnt4* (*Figure 4—source data 1*). We find that at ss4 and ss8 these gene modules are indeed upregulated in the forebrain/aPPR and midbrain/hindbrain, respectively (*Figure 4—figure supplement 1A*). To focus our analysis on the segregation of neural, neural crest and placodal cells rather than on A-P patterning, A-P restricted modules were removed. The remaining five modules show broadly pan-neural (GM9), pan-placodal (GM21, GM23, GM24) or neural crest cell (GM13) expression at ss4 and ss8 (*Figure 4E*).

## Latent time analysis suggests late emergence of neural crest cells

Given the cellular heterogeneity observed at early timepoints, discrete developmental stages do not necessarily accurately represent the developmental progression of individual cells towards a given fate. In contrast, latent time can measure the biological clock of individual cells as they differentiate and thus represents the order of cells along a developmental trajectory. To explore the expression dynamics of the five selected gene modules as neural, neural crest, and placodal fates emerge, we first ordered cells across latent time using scVelo for the entire dataset (*Figure 4C* and *Figure 4—figure supplement 1B*). To investigate the segregation of all three fates, three separate trajectories need to be distinguished. To do this, we calculated the probability of each cell to transit towards one of three terminal states (*Figure 4—figure supplement 1C*). These probabilities were obtained using CellRank (*Lange et al., 2022*), which leverages RNA velocity and transcriptomic similarity information to predict transitions between cells. Visualising the fate probabilities across the full dataset shows that cells with low latent time values (i.e. early cells) have low probabilities of becoming neural, neural crest or placodal cells (*Figure 4D*, see *Figure 4A* for stage reference). As cells 'age' the fate probability increases as they become transcriptionally similar to the terminal state. Unlike placodal and neural fates, neural crest fate probabilities are only high in 'late' cells (*Figure 4D*), reflecting the fact that defined neural crest cell clusters only appear at HH7.

## Gene module dynamics reveals key differences in the activation of neural, neural crest, and PPR programmes

Having predicted the developmental trajectory of cells using scVelo and CellRank, we next modelled the expression of the five selected gene modules as a function of latent time with the aim to characterise their dynamics. Gene expression dynamics were modelled using generalised additive models (GAMs). GAMs were chosen because gene expression is highly dynamic and does not change linearly over time. For each gene module, we fitted three separate GAMs, one for each fate (*Figure 4E*; left). The three fates were distinguished using the previously calculated fate absorption probabilities. This modelling approach reveals dynamic changes of gene expression. Three gene modules – GM9, GM13, GM23 – are initially inactive and are upregulated at different points across latent time in cells with neural, neural crest, and placodal character, respectively.

GM9 becomes upregulated in all neural clusters and includes neural markers like *Lmo1* and *Sox21* (*Figure 4E* 'Neural GM9', *Figure 4—source data 1*). Although the expression of different genes within this module correlates broadly, there is some gene-gene variation. For example, *Lmo1* is the first gene to be expressed and exhibits highly dynamic expression over time. It is first found at HH5 within the eCN, pNP and iNP, before being upregulated across the entire neural plate at HH7 and ss4. At ss8, *Lmo1* is specifically downregulated within the MB. In contrast, *Sox21* is first upregulated at HH5/6 and is consistently expressed throughout all neural cell clusters.

In contrast, modelling reveals that GM13 is upregulated in cells with neural crest identity. Indeed, it contains well known neural crest markers *Pax7, Snai2, Sox10,* and *Tfap2b* (*Figure 4E* 'NC GM13', *Figure 4—source data 1*) and is initially broadly activated in NPB and neural crest clusters at HH7. At ss4 and ss8, this module remains strongly expressed within neural crest and delaminating neural crest

clusters. Within GM13 individual genes are activated at different time points: *Pax7* is the first gene to be expressed in the pNPB at HH5, while *Sox10* starts to be expressed only from ss4 onwards.

Finally, modelling the expression dynamics of GM23 shows that it is activated early and becomes upregulated in cells with placodal character across latent time. This prediction is supported by the fact that GM23 is highly expressed in PPR clusters at ss8 and contains bona fide placodal genes like *Six1* and *Eya2* (*Figure 4E* 'Placodal GM23', *Figure 4—source data 1*). At HH4, most GM23 genes are not expressed, but are broadly activated at HH5 in PPR, non-neural and NPB clusters, where they remain active until becoming restricted to PPR cells at ss4. Comparing the onset of activation of these three gene modules across latent time reveals a clear temporal order with the placodal module being activated first, followed by the neural and neural crest gene modules.

In contrast to these modules, modelling of GM24 and GM21 shows that they are initially expressed in all three cell populations before being rapidly downregulated within neural crest and neural cells (*Figure 4E* 'Epiblast GM21' and 'Epiblast GM24'). Indeed, genes in GM24 are broadly expressed across the epiblast at HH4, and then become gradually restricted to PPR, non-neural and NPB clusters at HH6/7, and to PPR cells at ss4/8 (*Figure 4A*). GM21 displays similar dynamics but is restricted to non-neural clusters earlier at HH5. Although some genes within these epiblast modules have been described as non-neural/PPR specific based on in situ hybridisation (*Tfap2c*, *Bambi*, *Homer2*, *Tspan13*) (*Bell et al., 2004*; *Hintze et al., 2017*; *Mehdizadeh et al., 2021*; *Reichert et al., 2013*; *Rothstein and Simoes-Costa, 2020*), others are expressed within the early neural territory (*Epcam*, *Sall4*, *Cited4*) (*Barembaum and Bronner-Fraser, 2007*; *Bell et al., 2004*; *Trevers et al., 2023*; *Figure 4—source data 1*).

In summary, modelling the expression dynamics of selected gene modules provides new insight into the sequential activation of fate-specific developmental programmes, as well as establishing the order of molecular events as neural, neural crest and placodal cell fates develop. We show, using an unbiased gene module approach, that a placodal gene module is activated first, followed by neural and neural crest modules. While neural and neural crest modules are specifically activated in the relevant cell clusters alone, two modules that are later restricted to the PPR are initially broadly expressed across the epiblast. We therefore propose that to establish robust neural or neural crest transcriptional signatures, ectodermal cells first require the downregulation of epiblast modules.

## Spatially restricted co-expression of genes at the NPB

Dynamic modelling of gene expression reveals that although so-called NPB specifiers are initially co-expressed, they later become enriched in neural, neural crest or placodal cells (*Figure 5—figure supplement 1* ). Where are co-expressing cells located in the developing embryo? To investigate this, we modelled the expression of 'NPB specifiers' across the mediolateral axis, before validating their spatial expression in vivo using in situ hybridisation chain reaction (HCR).

We chose to model the spatial expression of 'NPB specifiers' using our HH7 scRNAseq data as this stage marks the start of placodal and neural crest segregation (*Figure 3A*). Principal component analysis reveals that HH7 cells are ordered along the principal component (PC) 1 based on their M-L position in vivo; PC1 therefore provides an axis through which we can model spatial gene expression (*Figure 5A*). Smoothed expression profiles across PC1 were modelled using GAMs calculated for each NPB specifier (*Figure 5B*).

Our models predict that *Msx1* and *Pax7* are upregulated medially relative to *Dlx6* and *Six1*, in line with their roles in neural crest and placodal specification, respectively (*Basch et al., 2006*; *Brugmann et al., 2004*; *Christophorou et al., 2009*; *McLarren et al., 2003*; *Monsoro-Burq et al., 2005*). Interestingly, *Pax7* and *Six1*, which are upregulated later than either *Msx1* and *Dlx6* (*Figure 3E*), are predicted to exhibit a smaller region of co-expression with each other than with either *Msx1* or *Dlx6* (*Figure 5C*). This highlights that *Pax7* and *Six1* are significantly enriched in neural crest and placodal progenitors, respectively. *Tfap2a*, which is required for both placodal and neural crest specification (*Bhat et al., 2013*; *Rothstein and Simoes-Costa, 2020*), is predicted to overlap extensively with both *Pax7* and *Six1*.

HCR at the same stage validates the predicted levels of co-expression across the M-L axis in vivo (*Figure 5C–E*). We quantified M-L expression by taking intensity measurements at three different A-P positions (*Figure 5C*; yellow horizontal lines). The intensity measurements for each gene were Z-scored across all three axial levels to allow for comparative relative measurements of gene expression

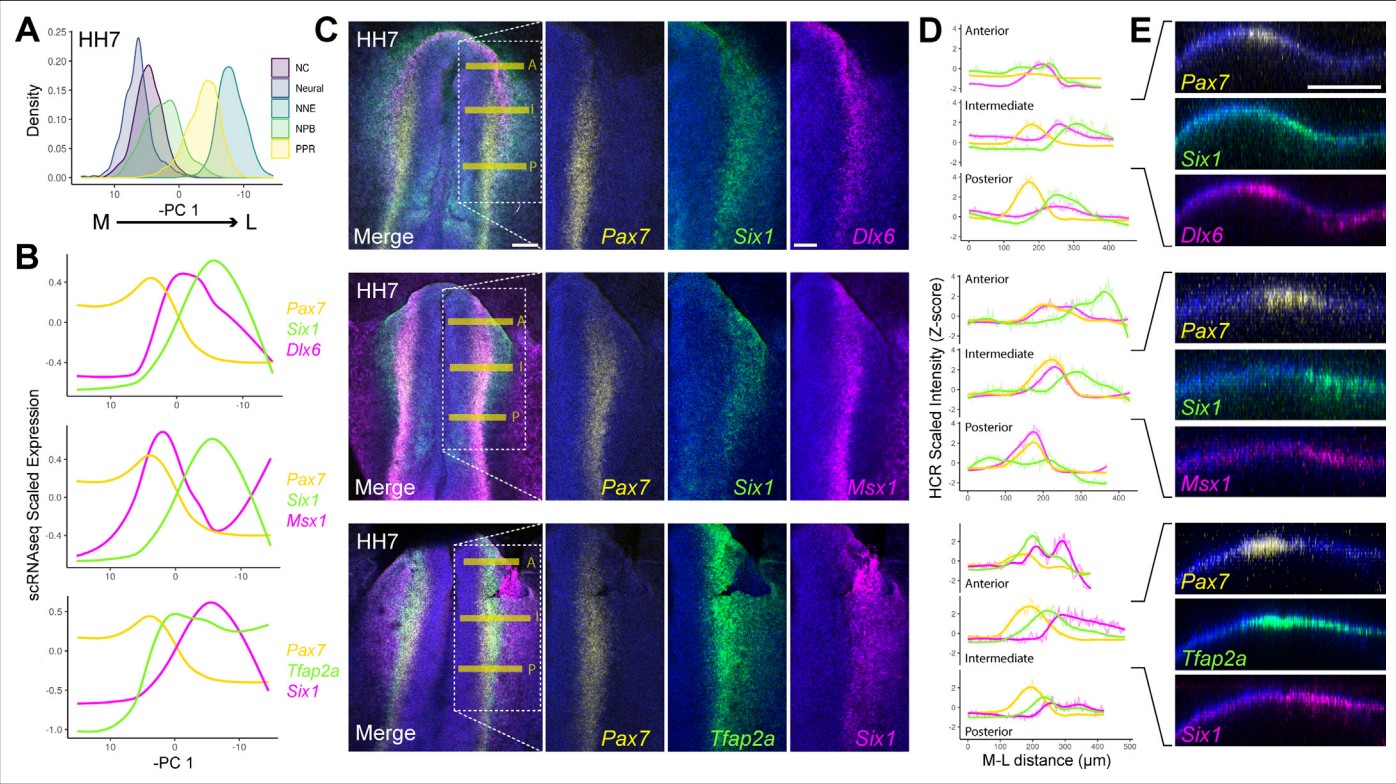

**Figure 5.** HCR validates spatial restriction of classical NPB specifiers. (**A**) Distribution of cells at HH7 from major ectodermal cell lineages across principal component 1 (PC 1), revealing medio-lateral patterning across this axis. The x-axis has been inverted to reflect the positioning of cell populations across the medio-lateral axis in vivo. (**B**) Spatial gene expression modelling of key placodal and neural crest specifiers at HH7 across the inverse of PC 1. Generalised additive models were fitted for each gene to predict their medio-lateral pattern of expression in the embryo. (**C**) Whole mount in situ hybridisation chain reaction (HCR) images at HH7 for combinations of markers modelled in B. Overlayed dotted box in the merged image show the region of interest displayed for each separated colour channel. (**D**) Fluorescent intensity measurements taken at anterior 'A', intermediate 'I' and posterior 'P' regions depicted by the yellow bars in C. Intensity measurements were scaled for each gene across the three axial levels to allow for relative comparisons between different axial regions. (**E**) Virtual crosssections taken from the intermediate region, highlighting expression of each marker within the embryonic ectoderm. Scale bars in C represent 100 μm (left image) and 50 μm (right image); they apply to all images in the same column. Bar in E represents 100 μm and applied to all sections.

The online version of this article includes the following figure supplement(s) for figure 5:

**Figure supplement 1.** Generalised additive models (GAMs) of average normalised gene expression for previously characterised neural plate border 'specifiers'.

(*Figure 5D*). We note a striking correlation in the expression intensity measurements taken from the intermediate axial level (*Figure 5D*; Intermediate) and the predicted M-L expression patterns from our scRNAseq data (*Figure 5B*), which represent an aggregate of all A-P regions.

Taken together, both our scRNAseq and HCR analysis reveal segregation of neural crest and placodal markers across the M-L axis, with considerable heterogeneity in gene co-expression depending upon the NPB markers in question. Importantly, we show that there are clear differences in their co-expression along the A-P axis. While *Dlx6* and *Six1* are strongly expressed anteriorly, *Pax7* and *Msx1* show increased expression levels posteriorly (*Figure 5D*). These findings are in line with the well characterised role of these transcription factors during development of placodal and neural crest fates, respectively (for review: *Grocott et al., 2012*; *Meulemans and Bronner-Fraser, 2004*; *Thiery et al., 2020*). Together, these results highlight that whilst the NPB can be defined as a territory adjacent to the developing neural plate, a unique NPB cell state cannot be defined transcriptionally using previously established criteria.

## BLUPs: multi-potent progenitors at the NPB?

Recent studies in other systems have revealed that undecided multi-potent progenitors often transition through an intermediate state characterised by high transcriptional heterogeneity and co-expression

of competing transcriptional programmes (*Soldatov et al., 2019*; *Subkhankulova et al., 2023*). Given the heterogeneity of cells at the NPB and the lack of a definitive transcriptional signature, we investigated whether cells at the NPB are indeed transcriptionally undecided.

## Co-expression of fate specific gene modules predict ectodermal cell fate choice

To assess the potential presence of multi-potent progenitors at the NPB, we identified gene modules that are restricted to either forebrain, mid-hindbrain, neural crest or placodal cells at late stages (ss8) and investigated their co-expression at earlier stages. Fifteen modules were differentially expressed in at least one cell state at ss8 (*Figure 6—figure supplement 1* and *Figure 6—source data 1*). GM9-13 and GM16-17 are characteristic for neural cells with different A-P identities. GM10 is upregulated in the forebrain and contains the canonical forebrain markers *Hesx1* and *Six3*. GM16 is enriched within mid-hindbrain clusters and expresses known mid-hindbrain markers *Sp5* and *Nkx6-2*; this module also exhibits low levels of expression in other neural clusters, but importantly, is downregulated in the neural crest. Seven gene modules are absent from neural populations but are expressed in placodal cells (GM4-6), neural crest (GM1-2) or both (GM7-8). GM5 contains well-characterised placodal markers including *Six1*, *Homer2*, and *Znf385c* and is expressed in both aPPR and pPPR clusters. This module was therefore selected as a pan-placodal module. GM2 is expressed in both neural crest and delaminating neural crest clusters and contains neural crest specifiers including *FoxD3*, *Snai2*, *Sox9*, *Pax7*, and *Msx1*, as well as the Wnt ligands *Wnt1* and *Wnt6*. GM2 thus represents a bona fide neural crest specific module. Although GM1 contains multiple neural crest markers (*Ets1*, *Sox10*, *Sox8*; *Figure 6—source data 1*), it is only expressed in delaminating neural crest cells and was therefore excluded for further analysis. Thus, GM10 (forebrain), GM16 (mid-hindbrain), GM5 (pan-placodal), and GM2 (pan-neural crest) modules were selected for subsequent co-expression analysis (*Figure 6*).

Our RNA velocity analysis at 4-ss (*Figure 3F and G*) predicts multiple branching events between ectodermal fates (placodal/forebrain, neural crest/placodal and mid-hindbrain/neural crest). To screen for putative multi-potent progenitors, we assessed the co-expression of gene modules later restricted to each fate in a pairwise manner (*Figure 6*). First, we scaled the average expression of selected gene modules and visualised their co-expression (placodal/forebrain, GM5/GM10; neural crest/placodal, GM2/GM5; mid-hindbrain/neural crest, GM16/GM2). To highlight cells with high levels of co-expression, we removed cells which do not express either gene module below a minimum heuristic threshold of 0.3 scaled expression.

At head process stages (HH5) almost all cells co-express all gene modules suggesting broad multi-potency. At later stages (ss4 and ss8) only small subsets of cells retain high levels of co-expression (*Figure 6C' and D'*, dashed boxes). This shift from broad to restricted co-expression suggests that ectodermal fate segregate between HH7 and ss8. However, the decrease in gene module co-expression does not occur evenly across the ectoderm. Visual inspection of the 4ss UMAPs highlights that cell co-expressing placodal/forebrain gene modules (*Figure 6D*) no longer co-express neural crest/mid-hindbrain gene modules (*Figure 6E*), while the reverse is true for cells at the neural crest/mid-hindbrain branch point. At the placodal/neural crest boundary, cells continue to co-express all four gene modules, albeit at different levels indicating that they may retain the potential to give rise to all four fates (*Figure 6C–E*). While individual cells at different axial levels clearly face different choices (*Bhattacharyya et al., 2004*; *Streit, 2002*; *Xu et al., 2008*), our analysis also points to a different order of cell fate restrictions. In summary, at the NPB cell fate decisions as revealed by co-expression of competing gene modules are highly heterogeneous. We term cells which co-express alternative fate gene modules 'Border-Located Undecided Progenitors' (BLUPs).

Overall, our co-expression analysis highlights that BLUPs are transcriptionally undecided and co-express gene modules that characterise alternative ectodermal fates. This expression profile is likely to endow them with the potential to give rise to different NPB fates. Thus, while the NPB demarcates an anatomical region surrounding the neural plate, it lacks a unique transcriptional signature. We propose that undecided NPB progenitors are better defined based on the co-expression of alternate gene modules. The proportion of these cells decreases over time, but a small population at ss4 retains this characteristic and is therefore likely to retain bi- or multipotency even as the neural tube closes. Further exploration is required to understand how the transcriptional heterogeneity at the NPB is resolved.

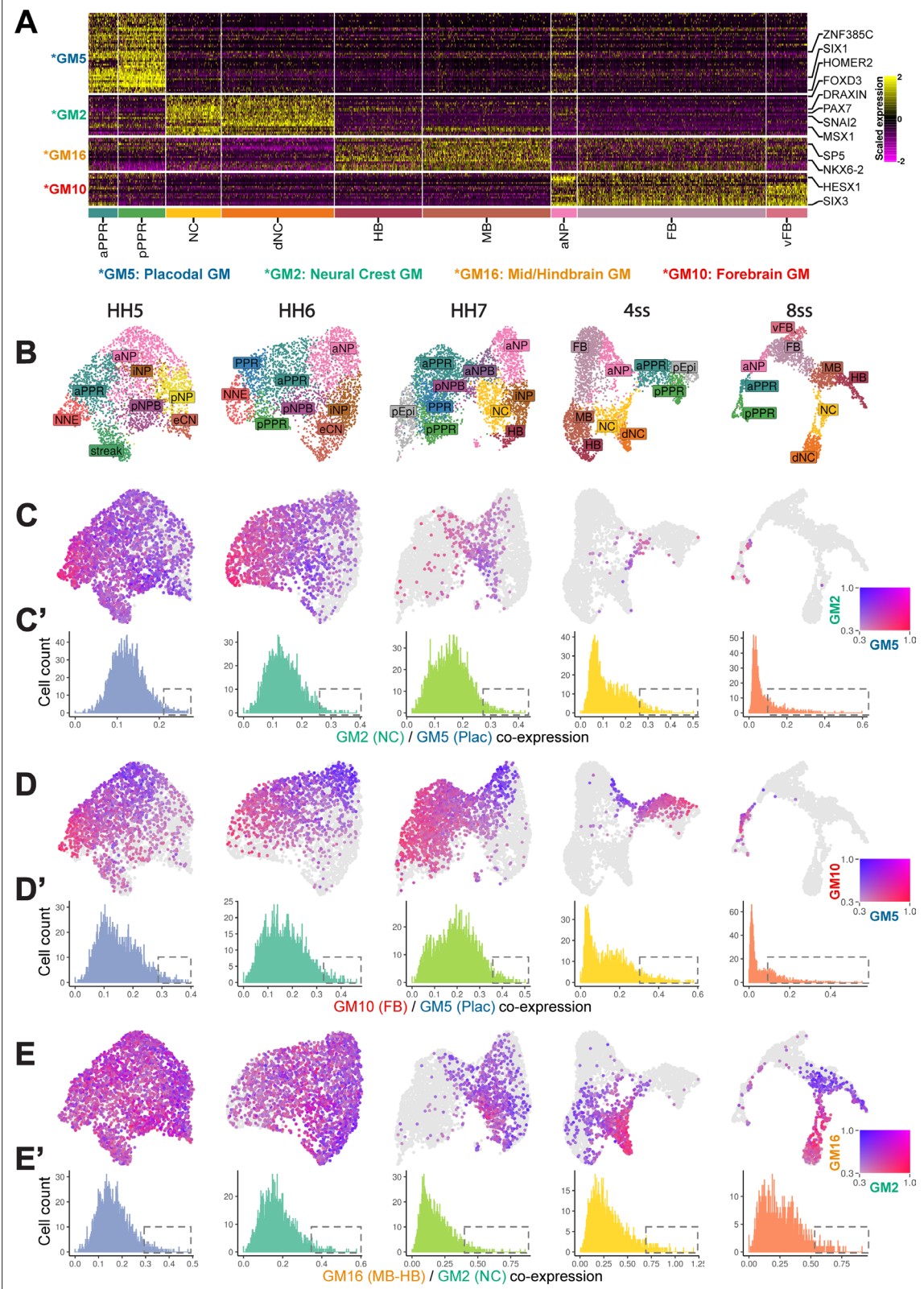

**Figure 6.** Border-Located Undecided Progenitors' (BLUPs) co-express placodal and neural crest gene modules. (**A**) Heatmap displaying pan-placodal (GM5), pan-neural crest (GM2), mid-hindbrain (GM16), and forebrain (GM10) gene modules at ss8 which have been subset from gene modules in *Figure 6—figure supplement 1A*. (**B**) UMAP plots from each developmental stage coloured by cell state. (**C**) Co-expression analysis of the pan-neural crest and pan-placodal modules (see Materials and methods) at each developmental stage. Cells which co-express both modules above 0.3 are

*Figure 6 continued on next page*

*Figure 6 continued*

coloured. (**C'**) Histograms revealing the distribution of co-expression (calculated as the product of gene module expression) of the pan-neural crest and pan-placodal modules at each developmental stage. At later stages (ss4-ss8), the distribution of co-expression shifts from a normal to a negative binomial distribution. Dashed boxes highlight the shift in the proportion of cells maintaining relative high co-expression at later developmental stages. (**D**) Co-expression analysis of pan-placodal (GM5) and forebrain (GM10) gene modules. Cells which co-express both modules above 0.3 are coloured. (**D'**) Histograms revealing the distribution of co-expression (calculated as the product of gene module expression) of pan-placodal and forebrain gene modules at each developmental stage. (**E**) Co-expression analysis of neural crest (GM2) and mid-hindbrain (GM16) gene modules. Cells which co-express both modules above 0.3 are coloured. (**E'**) Histograms revealing the distribution of co-expression (calculated as the product of gene module expression) of neural crest and mid-hindbrain gene modules at each developmental stage.

The online version of this article includes the following source data and figure supplement(s) for figure 6:

**Source data 1.** Gene lists from gene modules calculated at ss8 and filtered to include only those differentially expressed between cell states.

**Figure supplement 1.** Gene modules calculated from ss8.

## A temporal hierarchy of gene expression accompanies the emergence of definitive neural crest and placode precursors

To explore how BLUPs segregate towards neural crest and placodal fates, we selected NPB (aNPB, pNPB), neural crest (NC, dNC), and PPR cell clusters (aPPR, pPPR, PPR) from HH5-ss8. Visual UMAP analysis revealed that NPB cells are located at the boundary between neural crest and aPPR clusters, highlighting a bifurcation of neural crest and placodal fates between HH7 and ss4 (*Figure 7—figure supplement 1*). Focusing solely on the NPB subset provides us with a more detailed understanding of the gene expression hierarchy that could play a role in the segregation of placodal and neural crest fates.

To model gene expression dynamics during lineage segregation at the NPB, we performed scVelo analysis on the NPB subset (*Figure 7A*) and used CellRank to determine the probability that each cell will become neural crest or placodal (*Figure 7A–D*). We then identified putative lineage specifiers by calculating gene modules which are differentially expressed between placodes and the neural crest. These 10 differentially expressed gene modules were further filtered to include only those that contained genes expressed in neural crest, delaminating neural crest or PPR cell states (as defined in the binary knowledge matrix). This analysis identified three modules (GM12-14) upregulated in placodal and four in neural crest cells (GM40 and GM42-44) between HH7 and ss8 (*Figure 7E*, *Figure 7—source data 1*). Using our latent time and lineage probability measurements, we model the dynamics of these 7 gene modules during the segregation of the placodes and neural crest (*Figure 7E*; left).

Of the placodal modules, GM14 and GM12 are expressed as early as HH5. GM12 is expressed throughout both aPPR and pNPB and contains early non-neural/PPR markers like *Gata2/3* (*Figure 7E*, *Figure 7—source data 1*). At HH6, GM12 is downregulated in the NPB and remains confined to the PPR clusters until ss8. Unlike GM12, GM14 is initially restricted to the PPR and only activated in the pNPB at HH7. At ss4 and ss8 its expression is confined to the PPR, with a higher expression in anterior cells. This module contains pan-placodal genes *Dlx5/6* as well as anterior markers *Six3* and *Pax6*. GM13 is the last placodal module to be activated and contains the definitive placodal transcripts *Six1* and *Eya2*. It is first expressed at low levels at HH6 before being broadly activated across the PPR at HH7 through to ss8. We also observe weaker expression of this module at the NPB cells at HH7. To visualise the sequential activation of these modules across latent time, gene expression of each cell was weighted based on its placodal fate probability. This analysis shows that GM12 is activated before GM14 and GM13 in latent time. Whilst the expression of GM12 and GM14 does not change dramatically, GM13 is highly upregulated over latent time within the placodal cells (*Figure 7E* and *Figure 7—figure supplement 1B*).

The four neural crest gene modules each contain *bona fide* neural crest transcriptional regulators including *Msx1*, *Zic1*, *Zeb2*, *Lmx1b* (GM40), *Pax7*, *Snai2*, *Axud1*, *Foxd3*, *Tfap2b*, *Sox9* (GM42), *Sox5*, *Id1/2* (GM44) and *Sox10*, *Ets1*, *Lmo4* and *Sox8* (GM43) (*Figure 7E* and *Figure 7—source data 1*). Our analysis groups these factors unbiasedly into four modules; modelling their expression across latent time weighted by neural crest fate probability shows a clear sequential activation from GM40 to GM42, GM44, and GM43 (*Figure 7—figure supplement 1C*). GM40 is activated first, initially broadly across aPPR and pNPB clusters at HH5 and HH6, before becoming restricted to neural crest and NPB clusters from HH7 onwards. By ss8, it is downregulated in the delaminating neural crest population.

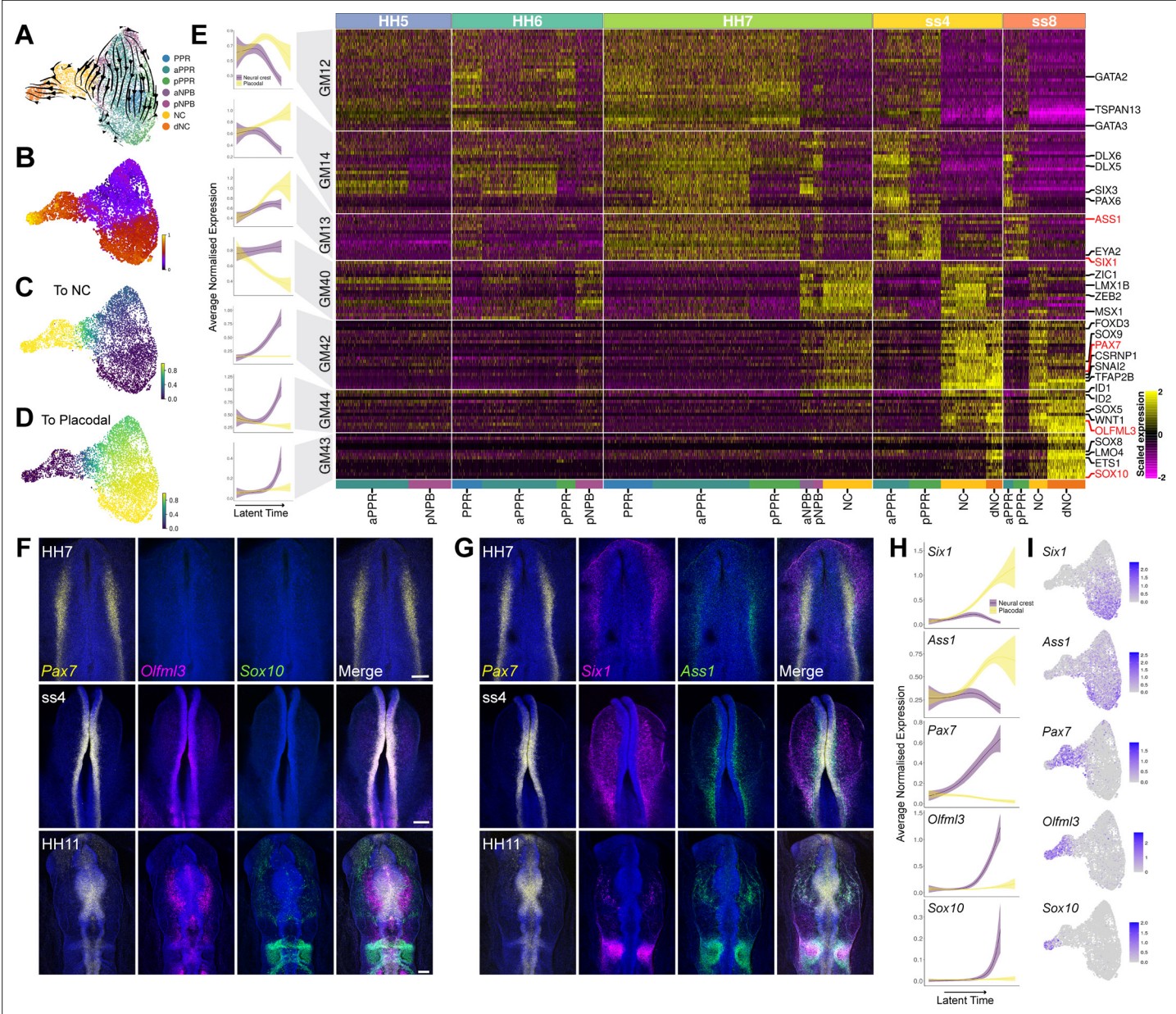

**Figure 7.** NPB gene module analysis reveals temporal hierarchy of gene expression during development of neural crest and placodes. (**A**) UMAP plot with cells coloured by cell state classifications and overlaid with RNA velocity vectors depicting the predicted directionality of transcriptional change. (**B**) UMAP plot of the NPB subset with cells coloured by latent time. (**C–D**) UMAP plots showing the fate absorption probabilities of each cell towards the specified terminal states (neural crest or placodes). (**E**) Left: Gene module dynamics plots displaying generalised additive models (GAMs) of average normalised gene module expression across latent time. GAMs are weighted by the fate absorption probability of each cell towards either placodal or neural crest terminal states. Shaded area represents upper and lower 95% confidence intervals. Right: Heatmap showing gene modules calculated across the NPB subset (full list of gene modules available in *Figure 7—source data 1*). To identify neural crest and placodal cell state signatures, gene modules were filtered to include those that show differential expression between placodal and neural crest cell states (see methods). Genes specified as neural crest or placodal in our binary knowledge matrix (*Supplementary file 1*) were selected as bait genes to further filter the gene module list for visualisation. Known and novel placodal and neural crest markers highlighted in red were validated by in situ hybridisation chain reaction (HCR) (**F–G**). (**F**) Whole mount in situ HCR at HH7, ss4 and HH11 validating the expression of *Olfml3* at the NPB and delaminating neural crest. (**G**) Whole mount in situ HCR at HH7, ss4 and HH11 validating the expression of *Ass1* in the pre-placodal region. (**H**) Gene dynamics displaying GAMs of average normalised gene expression across latent time. GAMs are weighted by the fate absorption probability of each cell towards either placodal or neural crest lineages. Shaded area represents upper and lower 95% confidence intervals. (**I**) Feature plots for genes validated by in situ HCR on the HH7 stage UMAP. Scale bars in G represent 100 µm; bars apply to all images in the same row.

The online version of this article includes the following source data and figure supplement(s) for figure 7:

*Figure 7 continued on next page*

*Figure 7 continued*

**Source data 1.** Gene lists from gene modules calculated in the NPB subset and filtered to include only those that show differential expression between cell states and not sequencing batches, and then further filtered to include only those containing genes that were defined within neural crest, neural crest or PPR cell states in the binary knowledge matrix.

**Figure supplement 1.** Gene modules from the NPB subset and their expression acorss latent time.

GM42 and GM44 are highly upregulated only in neural crest and delaminating neural crest cells, with GM42 first expressed at HH7 and GM44 at ss4. Lastly, GM44 is activated specifically within the delaminating neural crest cells at ss4 where it is further upregulated at ss8 (*Figure 7E*). This order suggests a hierarchical relationship between these factors in the neural crest gene regulatory network.

While gene module selection was based on prior knowledge of marker genes, this analysis also allows us to predict novel factors with similar expression profiles. We selected *Ass1* and *Olfml3* to validate the expression of novel placodal and neural crest markers, respectively, using HCR (*Figure 7F and G*). Like the placodal specifiers *Six1* and *Eya1, Ass1* is found in GM13. HCR confirms that *Ass1* and *Six1* are co-localised within the lateral NPB at HH7 and ss4, and in the otic placode at HH11 (*Figure 7G*). These experiments also reveal a greater degree in co-expression of *Pax7/Ass1* than *Pax7/Six1* (*Figure 7G*; HH7 and ss4). in line with our dynamic gene expression modelling, which predicts that *Ass1* is initially upregulated in both placodal and early neural crest cells before being downregulated in the neural crest (*Figure 7H,I*). *Olfml3* was chosen as a novel neural crest marker, as it is found in GM44 which is predicted to be upregulated within the neural crest in between GM42 (*Pax7*) and GM43 (*Sox10*) (*Figure 7H*). HCR confirmed the predicted sequential expression of these markers, with *Pax7* active in the posterior NPB at HH7, *Pax7*, and *Olfml3* co-expressed at NPB at ss4, and all three markers expressed within the neural crest at HH11 when the neural crest is undergoing delamination (*Figure 7F*).

In summary, analysing gene module dynamics across the NPB subset reveals the sequential activation of different co-regulated genes in placodal and neural crest cells. Many genes within these modules have been previously characterised and are known to form part of the neural crest or placode gene regulatory network (for review: *Grocott et al., 2012*; *Pla and Monsoro-Burq, 2018*; *Schlosser, 2006*; *Schlosser, 2014*; *Simões-Costa and Bronner, 2015*; *Thiery et al., 2020*). This unbiased approach groups these factors and predicts a hierarchical order, as well as identifying new candidate factors which may play important roles in the segregation of the neural crest and placodal fates.

## Discussion

The regulation of cell fate allocation at the NPB has been extensively studied in a range of model systems, giving rise to a wealth of knowledge regarding transcriptional regulators and their interactions. However, our understanding as to how individual cells undergo cell fate decisions has been limited by our inability to study transcription and gene regulation at a single cell level.

In this study, we characterise ectodermal cellular heterogeneity from primitive streak stages though to late neurulation in chick. We demonstrate that although transcriptional differences are apparent between cells at primitive streak stages, due to high levels of heterogeneity, cells do not cluster into discrete cell states until early neurulation (HH7). Instead, cell states are initially characterised by the overlapping expression of broad early neural and non-neural genes, as well as axial markers. We identify increased cell state diversification during neurulation, with neural, neural crest and placodal fates segregating from 1-somite to 8-somite stages. During this process subsets of cells continue to co-express neural, neural crest and placodal gene modules suggesting that they retain a broad developmental potential. Although single-cell lineage tracing and functional experiments are required to validate multipotency of cells at the NPB, the co-expression of competing gene modules appears to be characteristic of cells in an undecided state (*Soldatov et al., 2019*; *Subkhankulova et al., 2023*). We term these putative multipotent progenitors 'Border Located Undecided Progenitors' (BLUPs).

### Emergence of neural and neural crest cell signatures require downregulation of broad epiblast modules

Although specification networks for neural, neural crest and placodal cells are well established (for review: *Betancur et al., 2010*; *Grocott et al., 2012*; *Pla and Monsoro-Burq, 2018*; *Stundl et al.,*

*2021*; *Thiery et al., 2020*; *Trevers et al., 2023*), the order in which these lineages develop remains unclear. Our unbiased gene module analysis has identified groups of genes which are specifically upregulated within each cell type, broadly supporting these networks. *Pax7*, *Tfap2b*, *Snai2*, and *Sox10* cluster together and are upregulated in neural crest states, while *Six1*, *Eya2*, *Gata2/3*, and *Dlx5/6* are activated within placodal and *Lmo1* and *Sox21* in neural states. Furthermore, we uncover the order of molecular events as ectodermal cells diverge: placodal gene modules are activated first followed by neural and neural crest modules. These findings suggest that these cell states are emerge in that order. However, we also highlight developmental asynchrony between progenitors within each cell lineage. Although we classify placodal cells as early as HH5, transcriptionally undecided BLUPs can still be identified at ss4. These observations propose that cell fate allocation at the NPB is a gradual process, with the number of undecided progenitors decreasing over time as they commit to a given lineage.

Alongside characterising placodal, neural, and neural crest gene modules, we observe the expression of two modules which exhibit initial broad pan-epiblast expression at primitive streak stages, but which are confined to non-neural cell states at the 4 and 8-somite stages. Within these epiblast modules, unsurprisingly we note the expression of important placodal and non-neural genes *Tfap2c*, *Homer2* and the BMP inhibitor *Bambi*. In addition, we also find the pluripotency marker *Sall4* and *Tspan13*, with the latter shown to be positively regulated by Six1 (*Mehdizadeh et al., 2021*). A recent study in chick shows that a 'pre-border' state can already be defined prior to gastrulation based on transcriptional profiling, while in vitro experiments reveal that the pre-gastrula epiblast is specified as NPB. Once cells have acquired a 'pre-border' state they can become neural, neural crest and placodal precursors (*Trevers et al., 2018*). Given that the expression of epiblast modules becomes confined to the placodes later, we speculate that the development of the neural crest and neural fates requires the downregulation of the epiblast signatures prior to the activation of neural and neural crest modules.

## Co-expression of competing gene modules defines NPB cells

The NPB has largely been viewed as a unique territory where neural and non-neural gene expression overlaps and cells are defined by a distinct transcriptional signature. If this indeed is so, NPB markers should be uniformly expressed in NPB cells which subsequently give rise to neural and non-neural ectoderm, neural crest and placode precursors. Members of Zic, Tfap2, Pax, Msx, and Dlx transcription factor families have previously been identified as so-called NPB 'specifiers' based on functional experiments that showed them to be necessary for the formation of neural crest cells, placode precursors or both (for review: *Stundl et al., 2021*; *Thiery et al., 2020*). However, their expression is not uniform in the ectoderm surrounding the neural plate (*Basch et al., 2006*; *McLarren et al., 2003*; *Rothstein and Simoes-Costa, 2020*; *Streit and Stern, 1999*). For example, *Msx1* begins to be expressed at the posterior NPB and gradually expands anteriorly, while *Dlx5* expression is initially widespread and then becomes more highly expressed in the anterior NPB (*McLarren et al., 2003*; *Streit and Stern, 1999*). Our analysis extends these data and reveals that indeed none of the so-called NPB specifiers encompasses all cells located at the border of the neural plate and that the NPB cannot be defined by a unique transcriptional signature.

Here, we show that the NPB is an ectodermal territory surrounding the neural plate that contains cells with heterogeneous combinations of gene expression. These cells are characterised by the co-expression of gene modules which regulate the specification of alternative cell fates suggesting that they are undecided progenitors. Whether BLUPs are truly multipotent remains to be elucidated as do the molecular mechanism that control how conflicting transcriptional signatures are ultimately resolved.

While NPB cells at early stages can be defined by the co-expression of neural and non-neural modules with some subdivision along the anterior-posterior axis (*Figure 2*), cell states become more heterogeneous as the neural plate is firmly established. By the 1 somite stage, there is a clear heterogeneity that reflects potential fate choices along the anterior-posterior axis (*Figure 3*). Our co-expression analysis reveals that at the 4-somite stage, cells co-express placodal and forebrain modules, but not neural crest modules. In contrast, other cells co-express neural and neural crest, but not placodal modules, while a small subset of cells at the branching point between neural crest and placodes co-express modules characteristic for all three ectodermal fates (*Figure 6* and *Figure 6—figure supplement 1*). This suggests that not only are cells in the NPB transcriptionally heterogeneous with respect

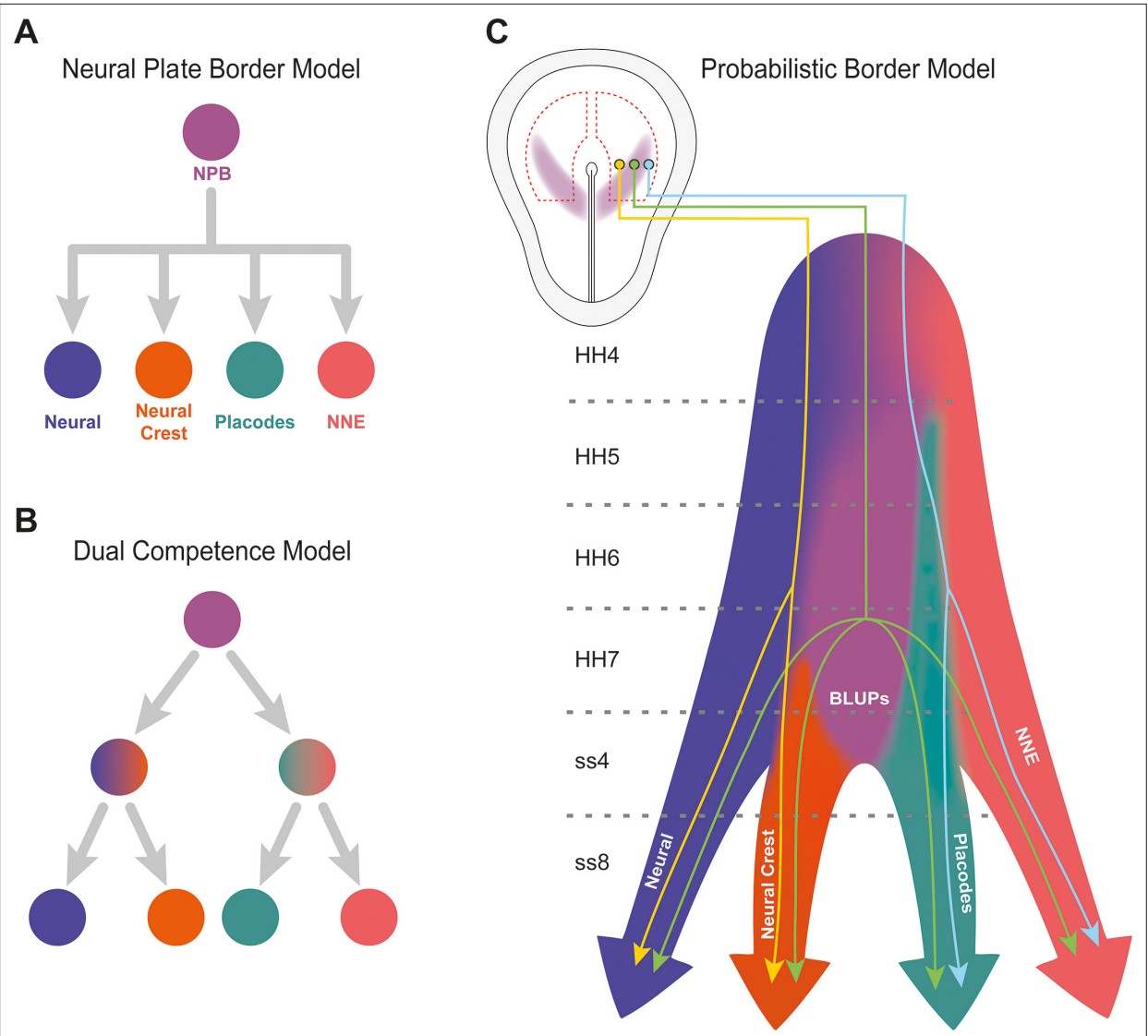

**Figure 8.** Probabilistic border model resolves conflicting models of lineage segregation at the neural plate border. (**A**) Schematic illustrating the neural plate border model. This model illustrates that cells at the neural plate border are multipotent and able to give rise to neural, neural crest, placodal, and non-neural ectoderm (NNE) lineages. (**B**) Schematic illustrating the dual competence model. This model suggests that the neural plate border is first defined into non-neural/placodal and neural crest/neural competence domains prior to further subdivision into each of the final four cell lineages. (**C**) Schematic illustrating the probabilistic border model. This model proposes a probabilistic model of cell fate allocation which is intrinsically linked to the spatiotemporal positioning of a cell. We suggest that cells located within the medial neural plate border (yellow cell lineage) will give rise to neural and neural crest, cells within the lateral neural plate border (blue cell lineage) to non-neural ectoderm and placodes, whilst subsets of cells (including Border Located Undecided Progenitors) will remain in an undecided state and continue to give rise to all four lineages even at late stages of neurulation (green cell lineage). Shaded background colours represent the cell state space.

to their medial-lateral position, but that multiple hierarchies of choices must exist depending on cell location along the anterior-posterior axis.

In summary, rather than considering the NPB as a unique territory of equivalent cells it should be viewed as a region of ectoderm surrounding the developing neural plate, where cells of undecided fates reside.

## A probabilistic model for cell fate allocation at the neural plate border

The 'binary competence' and 'neural plate border' models have provided useful frameworks for understanding cell fate determination during neural, neural crest and placodal segregation from

'undecided' precursors (*Figure 8A and B*) (for review: *Schlosser, 2006*; *Schlosser, 2014*; *Thiery et al., 2020*). Both models agree that the neural plate is initially surrounded by a territory containing progenitors for all ectodermal fates. The binary competence model suggests a specific sequence of progressive fate restriction shortly thereafter, with cells first becoming restricted to either neural (neural/neural crest) or non-neural (placodal/epidermal) fates (*Ahrens and Schlosser, 2005*; *Pieper et al., 2012*; *Maharana and Schlosser, 2018*). In contrast, the neural plate border model does not imply a specific order of fate decisions, but rather proposes that cells at the border remain multipotent and can differentiate into all ectodermal fates without suggesting a specific order of choices or binary choices (*Bronner-Fraser and Fraser, 1988*; *Roellig et al., 2017*; *Streit and Stern, 1999*). So far, the lack of experiments analysed at single cell resolution has precluded to resolve these apparently contrasting models. However, recent data from chick point to a transcriptional heterogeneity of NBP cells that cannot be accounted for in these models (*Roellig et al., 2017*; *Williams et al., 2022*). Our data provide a different perspective highlighting both cell state heterogeneity at the NPB and heterogeneity in the order and type of fate decisions encountered by NPB cells, as well as suggesting that fate decisions occur over a prolonged period and that spatial location of individual cells plays a critical role for their destination.

It is well established that cross-repressive interactions between transcription factors control cell fate choices at the NPB. For example, misexpression of the placodal factor Six1 promotes other placodal markers, while repressing neural and neural-crest-specific gene expression (*Brugmann et al., 2004*; *Christophorou et al., 2009*). Likewise, misexpression of Sox2 or Pax7 can shift proportional assignment of neural and neural crest fates (*Roellig et al., 2017*). Indeed, we find that not only individual genes previously thought to be 'fate specific', but whole gene modules overlap widely at early developmental stages (*Figure 6*) with many individual cells co-expressing markers for all ectodermal derivatives until at least head fold stages. While the number of such cells decreases over time, they continue to be found at predicted lineage branchpoints at ss4. It is widely assumed that co-expression of competing transcriptional programmes indicates that cells have not undergone a terminal fate decision remaining 'undecided' (*Soldatov et al., 2019*; *Subkhankulova et al., 2023*). The fact that BLUPs can still be identified at neurulation stages (see also *Roellig et al., 2017*) suggests that cell fate allocation at the NPB is a long drawn-out process that does not occur simultaneously in all cells.

In *Xenopus*, transplantation experiments carried out at definitive neural plate stages proposed that the neural plate has lost competence to generate placode precursors, while the ectoderm has lost the ability to produce central nervous system (*Ahrens and Schlosser, 2005*; *Pieper et al., 2012*). These experiments have led to the dual competence model, which suggests that by this stage the ectoderm is subdivided into neural/neural crest versus placode/epidermal domains and that this represents the first step in ectodermal fate segregation. In support of this, we show that around the 1 somite stage ectodermal cells fall into two 'superclusters' with neural/neural crest and placode/epidermal signatures (*Figure 3A*). However, we also find that cells continue to co-express neural, neural crest and placodal modules at mid-hindbrain levels even at the 4ss. Thus, even long after competence is supposedly restricted, cells continue to be transcriptionally undecided suggesting that there is no strict segregation. In addition, we observe that at forebrain levels, where no neural crest cells arise, individual cells co-express neural and placodal modules arguing against a strict restriction of neural and non-neural fates by neural plate stages.

While the persistence of undecided progenitor cells at late developmental stages supports the neural plate border model, our single-cell RNAseq data together with our spatial analysis of gene expression suggests a more complex scenario. Modelling and measuring gene expression across the medial-lateral axis of the NBP reveals graded co-expression of gene modules and individual transcription factors, respectively. Cells at the medial edge of the NPB express higher levels of neural crest modules or factors, while those at the lateral edge express higher levels of placodal modules or factors. In contrast, cells located in the middle of the NPB generally co-express intermediate levels of both. We therefore propose a probabilistic model for cell fate allocation at the NPB which considers levels of gene expression and the position of cells (*Figure 8*). According to this model medial NPB cells are more likely to generate neural crest and neural cells while lateral NPB cells have the propensity to produce epidermis and placodal derivatives. In contrast, cells in the centre of the NPB retain multipotency and continue to be in an undecided state with equal possibility to give rise to all four ectodermal derivatives. Our probabilistic model does not argue in favour of the dual competence

or neural plate border model but highlights cell state heterogeneity and the importance to consider timing and spatial location. Ultimately, we will need to combine single-cell sequencing and lineage tracing to evaluate different modes of fate allocation.

# Materials and methods
## Data collection and alignment
### Single cell dissociation and 10X single-cell mRNA sequencing

Fertilised chicken (*Gallus gallus*) eggs (Stewart, Norfolk UK) were incubated at 38 °C for 24–32 hr depending upon the developmental stage collected. Embryos were pinned and had the endoderm and mesoderm removed using dispase (10 mg/ml) before dissecting the ectodermal region of interest (*Figure 2A*). Multiple embryos from the same stage were pooled prior to dissociation. Samples were then dissociated in 250 µl pre-heated FACSmax cell dissociation solution (Amsbio, T200100) with papain (30 U/ml; Sigma, P3125) for 20 min at 37 °C. Samples were gently pipetted every 5 min to facil-itate dissociation. After dissociation, 250 µl resuspension solution was added to prevent any further dissociation (HBSS; non-acetylated BSA) (1 mg/ml; Invitrogen, 10743447); HEPES (0.01 M); non-essential amino acids (1 x; Thermo Fisher Scientific, 11140050); rock inhibitor Y-27632 (10 µM; Stem-cell Technologies). Cells were passed through a 20 µm cell strainer (Miltenyi Biotech, 130-101-812) to remove any debris, before pelleting and resuspending in 500 µl resuspension buffer. To remove any dead or dying cells as well as any remaining doublets, 1 µl 0.1 mg/ml DAPI was added to the cell suspension before FAC sorting. Given the low number of cells obtained from each embryo, samples for each stage were collected over multiple days and stored in 90% MeOH with 10% resuspension solution after FAC sorting. Prior to sequencing, samples from the same stage were pooled and resus-pended in DPBS with 0.5% non-acetylated BSA and 0.5 U/µl RNAse inhibitor (Roche 3335399001).

Cells underwent single-cell mRNA sequencing at the Francis Crick Institute, London. Library prepa-ration was carried out using the 10x3' single-cell mRNA v3 kit (10 x Genomics), before sequencing on the HiSeq 4000 (Illumina) to a target depth of 50 k reads/cell. Samples were sequenced in two batches (batch 1: HH4 (55 embryos), HH6 (23), ss4 (5), ss8 (4); batch 2: HH5 (19), HH6 (23), HH7 (20), ss4 (4)), with two stages (HH6 and ss4) sequenced in both batches (technical replicates) to allow for the correc-tion of any potential batch effects. Data from the first batch has previously been used for the charac-terisation of neural induction (*Trevers et al., 2023*) (ArrayExpress accession number: E-MTAB-10408).

### Read alignment

A custom pipeline was developed in Nextflow (v20.07.1) (*Di Tommaso et al., 2017*) to handle all GTF processing and read alignment whilst maintaining full reproducibility.

Prior to scRNAseq alignment, the GalGal6 ensembl 97 GTF file was modified to prefix genes from the Z and W sex chromosomes with their respective chromosome IDs (Z- and W-). Furthermore, we noticed several mitochondrial genes in the GTF were missing their chromosome ID, in turn preventing accurately calculating cell percentage mitochondrial content downstream; mitochondrial genes were therefore also prefixed in the GTF (MT-). Finally, we annotated the key neural crest specifier Snai2 in the GTF (ENSGALG00000030902). Reads were processed and aligned to GalGal6 using CellRanger 4.0.0 (10x Genomics).

The aligned reads from CellRanger (BAM) were re-counted using Velocyto (*La Manno et al., 2018*) to generate spliced and unspliced counts for each gene, which are used in subsequent RNA velocity analysis.

## Downstream pre-processing

All downstream analysis was wrapped into a custom Nextflow pipeline to simplify re-running the anal-ysis by handling parallel processing of data subsets.

### Data filtering

Quality control, data filtering, cell clustering, dimensionality reduction and initial data visualisation was carried out using Seurat v4.0.5 (*Hao et al., 2021*). First, genes expressed in fewer than five cells were removed from the dataset. Next, to ensure the removal of poor-quality cells whilst also minimising over-filtering, we first applied a medium filtering threshold (cells expressing fewer than 1000 genes,

more than 6500 genes, or had more than 15% mitochondrial content were excluded from subsequent analysis) followed by the removal of remaining poor-quality cell clusters.

### Dimensionality reduction and clustering

Linear dimensionality reduction was carried out using principal component analysis (PCA). The number of principal components (PCs) used to construct the subsequent shared-nearest-neighbour (SNN) graph was determined unbiasedly, using an approach developed by the Harvard Chan Bioinformatics Core (https://hbctraining.github.io/scRNA-seq/lessons/elbow_plot_metric.html). First, we calculate the PC at which subsequent PCs contribute less than 5% of standard deviation, and the point where the cumulative variation of PCs exceeds 90%. The larger of these two values is selected. Second, we calculate the PC where the inclusion of consecutive PCs contributes less than 0.1% additional variation. The smaller value from these two steps is selected as a cut-off.

Louvain clustering was carried out with varying resolutions depending upon the data subset in question. A high clustering resolution (resolution = 2) was used to identify poor quality clusters and contaminating cell states. This was to ensure that only poor-quality cells were filtered from the dataset. All other clustering was carried out using a resolution of 0.5.

### Filtering of poor-quality clusters

Cell quality was measured using unique gene and UMI counts. Clusters in which both metrics fell below the 25th percentile relative to the rest of the dataset were considered poor quality and excluded from further analysis.

### Data integration

After the removal of poor-quality cells, cells from the two separate batches were integrated into a single full dataset (*Figure 2—figure supplement 1H–I*). To avoid over-correction and removal of stage differences, we integrated the data from the two sequencing batches using STACAS v1.1.0 (*Andreatta and Carmona, 2021*). This R package allows for alignment of datasets containing partially overlapping cell states.

### Regressing out confounding variables

Before downstream analysis, we regressed out percentage mitochondrial content, cell sex, and cell cycle. After regressing out percentage mitochondrial content, we identified a residual sex effect whereby cells clustered according to the expression of either W or Z chromosome genes. This resulted in a duplication of cell clusters, most apparent at HH4 (*Figure 2—figure supplement 1A–D*). Cells were k-means clustered according to their expression of W chromosome genes and classified as either male or female. This classification was used to regress out cell sex. Cell cycle was observed to have a strong effect on cell clustering and was therefore also regressed out during data scaling (*Figure 2—figure supplement 1E*).

### Removal of contaminating clusters

We found small contaminating clusters of primordial germ cells, blood islands, mesoderm, and endoderm within the dataset (*Figure 2—figure supplement 1F,G*). To semi-unbiasedly remove these clusters, we calculated the average expression of candidate markers for each of these cell states and removed clusters in which the median expression of any of these modules was greater than the 90th percentile relative to the rest of the dataset (*Figure 2—figure supplement 1F''*).

## Downstream analysis

### Unbiased cell classification

Cell states were classified on each stage independently using a binary knowledge matrix (*Supplementary file 1*). First, the full dataset was split by stage and then re-scaled and re-clustered as described above. Our binary knowledge matrix consists of binarized expression of 76 genes across 24 cell states. For each gene, published in situ hybridisation expression patterns were used to determine expression within each defined cell state. Given that gene expression can vary dramatically between stages, and some cell states are known to not be present at specific timepoints (i.e.

delaminating neural crest at HH4), we restricted the possible cell states at specific stages. At HH4, there were seven possible cell states, including non-neural ectoderm (NNE), node, streak, extra-embryonic (EE), early NPB (eNPB), early neural (eN), early caudal neural (eCN). At HH5, there were 17 potential states, this included all cell states in the binary knowledge matrix, except for the later counterparts to the earlier cell states which are ventral forebrain (vFB), forebrain (FB), midbrain (MB), hindbrain (HB), neural crest (NC), delaminating neural crest (dNC) and presumptive epidermis (pEpi). At HH6, we included the same cell states as HH5 except for EE and eNPB given that no cell clusters were classified as such at HH5. For stages HH7, ss4 and ss8 we included 17 cell states, excluding the following states which are only found at earlier stages (NNE, node, streak, eN, eCN, eNPB, EE).

For each cell, we calculated the average scaled expression of the genes expressed in each cell state according to our binary knowledge matrix. Cell clusters were then assigned a cell state, depending on which cell state exhibited the highest median expression in that given cell cluster. Assigned cell states at each of the six sampled developmental stages were then transferred to the full dataset.

## Gene module identification and filtering

To calculate groups of highly correlated genes, we first filtered genes which do not correlate (Spearman correlation <0.3) with at least three other genes. Then we calculated gene modules using the Antler R package (*Delile et al., 2019*), which iteratively clusters genes hierarchically based on their gene-gene Spearman correlation. After each iteration, gene modules were filtered to remove gene modules which are not highly expressed (expressed in fewer than 10 cells) or which do not show consistent high expression across genes (fewer than 40% of genes in the module are expressed after binarization). Gene modules were calculated on different subsets of cells: the full dataset (*Figure 4E* and *Figure 4—figure supplement 1A*), ss8 (*Figure 6A* and *Figure 6—figure supplement 1A*), and the NPB subset (*Figure 7E*). Calculating gene modules on different subsets allowed us to zoom in on specific aspects of ectodermal lineage segregation.

Many genes correlate across the dataset, but do not show differential expression between cell states and therefore do not provide useful information regarding lineage segregation. Therefore, to identify which gene modules correlate with lineage segregation, modules were filtered to only include those in which more than 50% of their genes were differentially expressed (logFC >0.5, adjusted p-value <0.001) in at least one cell state relative to the rest of the dataset. For the NPB subset, this filtering of differential expression was just between the neural crest (NC, dNC) and placodal (aPPR, pPPR, PPR) states rather than between all cell states (logFC >0.25, adjusted p-value <0.001).

To remove gene modules which correlate with technical variation, we filtered those in which more than 50% their genes showed differential expression between sequencing batches (logFC >0.25, adjusted p-value <0.001).

For the gene modules calculated on the full dataset, we were particularly interested in gene modules that segregated into one of the three lineages (neural, neural crest or placodal) at later stages. To focus our analysis on these gene modules, we applied an additional filtering step to only keep gene modules in which more than 50% of genes were differentially expressed between one of the three lineages at ss8 (logFC >0.25, adjusted p-value <0.001). After filtering 9 gene modules remained (*Figure 4—figure supplement 1A*, *Figure 4—source data 1*). Modules were further manually filtered for visualisation (*Figure 4E*).

For the gene modules calculated on the NPB subset we wanted to focus on the co-expression and sequential activation of gene modules that contain well-characterised neural crest and PPR markers. We therefore further filtered NPB gene modules to include only those that had genes that were defined within neural crest, delaminating neural crest or PPR cell states in the binary knowledge matrix. After this final filtering step, 7 gene modules remained (*Figure 7E*, *Figure 7—source data 1*).

## Co-expression visualisation

The average normalised expression of each gene module was scaled between 0 and 1. A 100x100 two-colour matrix was created using Seurat's BlendMatrix function. Each cell was assigned a colour from this matrix based on their scaled average expression of the gene modules (*Figure 6C, D and E*). Only cells which express both modules above a threshold of 0.3 after scaling were visualised.

### Co-expression distribution plots

To investigate the distribution of co-expression between cells, co-expression histograms were plotted (*Figure 6C', D' and E'*). Co-expression values were calculated by multiplying the average normalised expression for each pair of gene modules.

### RNA velocity

Spliced and unspliced matrices obtained from Velocyto (*La Manno et al., 2018*) (see *Read Alignment*) were analysed using scVelo (*Bergen et al., 2020*) to generate cell velocity vectors. Velocyto matrices were intersected with each of the data subsets generated in Seurat. First and second order moments were then calculated for each cell across its 20 nearest neighbours using the top 20 PCs. RNA velocity was ran using the dynamical model whilst accounting for differential kinetics; this allows for differential splicing rates between genes as well as differential splicing kinetics between cell states.

Latent time was calculated using scVelo (*Bergen et al., 2020*) with the diffusion weighting parameter set to 0.2. When calculating latent time, the root cell was calculated from cells belonging to the earliest timepoint present in the data; when analysing cells from individual stage subsets, no rooting population was specified. For all latent time estimates, we used a diffusion weighting of 0.2.

### Modelling gene expression dynamics

Lineage inference was calculated using CellRank, which uses a probabilistic approach to assign cell lineage (*Lange et al., 2022*). Cell transition matrices were calculated using RNA velocity (0.8 weighting) and nearest neighbour graph (0.2 weighting) kernels. A softmax scale of 4 was set to prevent transitions tending towards orthogonality. When predicting cell transitions, terminal states were specified according to the lineages which we were wanting to model gene dynamics across. For the full dataset, we predicted cell transitions towards either neural, placodal or neural crest lineages. Terminal states were specified as: delaminating neural crest and neural crest cell states at ss4 and ss8 for the neural crest lineage; hindbrain, midbrain, and forebrain cell states at ss4 and ss8 for the neural lineage; anterior PPR, posterior PPR and PPR cell states at ss4 and ss8 for the placodal lineage (*Figure 4—figure supplement 1C*). For our NPB subset, we predicted cell transitions towards either placodal or neural crest lineages, with terminal states specified as: delaminating neural crest and neural crest cell states at ss8 for the neural crest lineage; aPPR and PPR cell states at ss8 for the placodal lineage. The output of CellRank was the probability that any given cell would differentiate towards any of the terminal states. The lineage absorption probability is represented as a decimal between 0 and 1, with the sum of all lineage absorption probabilities for any given cell equal to 1.

Gene expression dynamics were calculated by fitting generalised additive models (GAMs), which modelled average normalised gene expression as a function of latent time using the mgcv package in R. Negative binomial distributions were fitted using a log link function and 95% confidence intervals were calculated for visualisation. Lineage absorption probability obtained from CellRank were used to weight a GAM for each of the predicted lineages. To generate a smoothed prediction of gene expression change, a cubic spline was fitted with knots = 4. When modelling the dynamics of entire gene modules, all genes within a given module were included in the GAM.

### Modelling mediolateral gene expression patterns from scRNAseq data

Following principal component analysis of cells at HH7, the order of cell states across the inverse of PC 1 were found to reflect the M-L positioning of cells in vivo. Genes which were. To model the expression of placodal and neural crest markers across the M-L axis, GAMs were fitted for the scaled expression of each gene across PC 1 using ggplot2::geom_smooth.

### **Fluorescent in situ hybridisation chain reaction**

Hybridisation chain reaction v3 (HCR) was carried out as previously described (*Buzzi et al., 2022*; *Choi et al., 2018*). Embryos were fixed in 4% PFA for 1 hr at room temperature, dehydrated in a series of methanol in PBT, and stored overnight at –20 °C. Samples were rehydrated and treated with proteinase-K for 3 min (20 mg/mL). Next samples were post-fixed in 4% PFA for 20 min and sequentially washed on ice for 5 min in PBS, 1:1 PBT/5×SSC (5×sodium chloride sodium citrate, 0.1% Tween-20), and 5×SSC. Samples were then pre-hybridised in hybridisation buffer for 5 min on ice, followed by

30 min at 37 °C. Next, samples were hybridised overnight at 37 °C with HCR probes in hybridisation buffer (4 pmol/mL). Excess probe was washed off with 15 min washes in probe wash buffer at 37 °C, before preamplification in amplification buffer at room temperature for 5 min. Hairpins (30 pmol) were incubated at 95 °C for 90 s and left to cool to room temperature for 30 min before being added to the amplification buffer. Samples were then incubated with the hairpin/amplification buffer solution overnight at room temperature. Excess hairpins were washed off with two 5 min and two 30 min washes in 5×SSC. After a 5 min incubation in DAPI (10 mg/mL), samples were washed 3 times for 10 min in 5×SSC before being imaged using a Leica SP5 laser scanning confocal inverted microscope using the LAS AF software.

Intensity measurements were calculated by taking the sum of slices from the image Z-stacks taken at three anteroposterior levels in ImageJ. The three intensity measurements were then independently scaled (Z-score) for each gene to allow relative measurements of gene expression across the anteroposterior axis. Transverse virtual sections across the anteroposterior axis were obtained using the Imaris 9.3.1 software. 10 µm sections were generated with the orthogonal slice tool.

## Acknowledgements

We thank Teresa Rayon for assistance with establishing the single cell dissociation protocol and the Advanced Sequencing Facility at the Francis Crick Institute for the single cell RNA sequencing. We would also like to thank Igor Adameyko and Artem Artemov for guidance on co-expression analysis, Chantal Hubens for technical support, Alessandra Vigilante and Sami Leino for comments on the manuscript, and Streit and Luscombe lab groups for helpful discussions. This work was supported by the Wellcome Trust (108874/B/15/Z), the BBSRC (BB/S005536/1; BB/R006342/1) and in part by the Francis Crick Institute which receives its core funding from Cancer Research UK (FC001051), the UK Medical Research Council (FC001051), and the Wellcome Trust (FC001051). For the purpose of Open Access, the author has applied a CC BY public copyright license to any Author Accepted Manuscript version arising from this submission.

## Additional information

### Funding

| Funder | Grant reference number | Author |
|---|---|---|
| Biotechnology and Biological Sciences Research Council | BB/S005536/1 | Alexandre P Thiery<br>Ailin Leticia Buzzi<br>Andrea Streit |
| Biotechnology and Biological Sciences Research Council | BB/R006342/1 | Alexandre P Thiery<br>Ailin Leticia Buzzi<br>Andrea Streit |
| Wellcome Trust | 108874/B/15/Z | Nicholas M Luscombe |
| Wellcome Trust | FC001051 | Chris Cheshire<br>Nicholas M Luscombe<br>James Briscoe |
| Cancer Research UK | FC001051 | Chris Cheshire<br>Nicholas M Luscombe<br>James Briscoe |
| Medical Research Council | FC001051 | Chris Cheshire<br>Nicholas M Luscombe<br>James Briscoe |
| Wellcome Trust | 108874/Z/15/Z | Eva Hamrud |

The funders had no role in study design, data collection and interpretation, or the decision to submit the work for publication. For the purpose of Open Access, the authors have applied a CC BY public copyright license to any Author Accepted Manuscript version arising from this submission.

## Author contributions
Alexandre P Thiery, Conceptualization, Resources, Data curation, Software, Formal analysis, Validation, Investigation, Visualization, Methodology, Writing – original draft, Writing – review and editing; Ailin Leticia Buzzi, Conceptualization, Formal analysis, Validation, Investigation, Visualization, Methodology, Writing – original draft, Writing – review and editing; Eva Hamrud, Formal analysis, Investigation, Visualization, Methodology, Writing – review and editing; Chris Cheshire, Data curation, Software, Methodology; Nicholas M Luscombe, Supervision, Methodology, Writing – review and editing; James Briscoe, Supervision, Funding acquisition, Writing – review and editing; Andrea Streit, Conceptualization, Supervision, Funding acquisition, Writing – original draft, Project administration, Writing – review and editing

## Author ORCIDs
James Briscoe ⬤ http://orcid.org/0000-0002-1020-5240
Andrea Streit ⬤ http://orcid.org/0000-0001-7664-7917

## Decision letter and Author response
Decision letter https://doi.org/10.7554/eLife.82717.sa1
Author response https://doi.org/10.7554/eLife.82717.sa2

---

# Additional files

## Supplementary files
• Supplementary file 1. Binary knowledge matrix used for unbiased cell state classification. This matrix was established by binarizing the expression of 76 genes across 24 putative cell states based on in situ hybridisation expression patterns obtained from the literature.

• MDAR checklist

## Data availability
10x single cell RNAseq was carried out in two batches and is available under two separate accession numbers (ArrayExpress: E-MTAB-10408 and E-MTAB-1144). Our NGS alignments and downstream analysis have been wrapped into custom Nextflow pipelines allowing for full reproducibility. For the code used in this analysis, including links to our Docker containers, see our GitHub repository at https://github.com/alexthiery/10x_neural_plate_border (copy archived at *Thiery and Hamrud, 2023*). Finally, we have developed a user friendly ShinyApp to allow public exploration of our single cell RNAseq data at https://shiny.crick.ac.uk/thiery_neural_plate_border/.

The following datasets were generated:

| Author(s) | Year | Dataset title | Dataset URL | Database and Identifier |
|---|---|---|---|---|
| Thiery AP, Buzzi AL, Hamrud E, Cheshire C, Luscombe NM, Briscoe J, Streit A | 2022 | NPB dataset 2 | https://www.ebi.ac.uk/biostudies/arrayexpress/studies/E-MTAB-1144 | ArrayExpress, E-MTAB-1144 |
| Thiery AP, Buzzi AL, Hamrud E, Cheshire C, Luscombe NM, Briscoe J, Streit A | 2023 | A gene regulatory network for neural induction | https://www.ebi.ac.uk/biostudies/arrayexpress/studies/E-MTAB-10408 | ArrayExpress, E-MTAB-10408 |

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
