## [Editor Report]

This is a useful study that describes the heterogeneous expression of a very large number of genes ostensibly associated with the earliest cell fate decisions in the ectoderm of chicken embryos. The evidence is solid, and the authors mostly succeed in presenting their complex strategy of single cell RNA-seq and subsequent data analysis in a compelling way. However, some of the conclusions about the ultimate fates of the differentiating cells remain speculative and any further validation would require functional studies and experimental tests of hypotheses. Overall, the methods applied, and the data generated by this study will be valuable to developmental biologists, especially those who study patterning at the neural plate border in vertebrates.

---

## [Decision Letter]

**Decision letter after peer review:**

Thank you for submitting your article "A gradient border model for cell fate decisions at the neural plate border" for consideration by *eLife*. Your article has been reviewed by 3 peer reviewers, and the evaluation has been overseen by a Reviewing Editor and Kathryn Cheah as the Senior Editor. The following individual involved in the review of your submission has agreed to reveal their identity: Gerhard Schlosser (Reviewer #2).

Essential revisions:

1. The Reviewers had major concerns about the language used to describe your results, your terminology, the interpretations of your findings, and the claims made about the significance of your study. The Reviewers felt that for your claims to be substantiated as written, you would have to perform additional proof-of-concept experiments and confirmatory analyses. Theoretically, these would include new gain- and loss-of-function experiments, lineage tracing analyses, and heterotopic transplantations.

However, given that your submission constitutes a Tools and Resources manuscript and not a Research Article, we are not asking you to perform additional experiments. Instead, the Reviewers are asking you to take very seriously all their other their requests (major and minor) for a substantial revision of the abstract and text, and a reanalysis of some of the data as detailed and described in each of their Recommendations sections.

2. Specifically, we are asking you to revise your manuscript to address what the Reviewers identify as conceptual problems with the interpretation of your data, unsubstantiated claims about the significance of your results, insufficient discussion of other similar previous studies/lineage tracing experiments, and highly speculative conclusions. For example, the Reviewers take issue with your use of the term BLUPs for transcriptionally unstable cells and your suggestion of a gradient model that they feel lacks molecular evidence. The Reviewers have also asked you to define and use terms like commitment, specification, etc. with more precision. Other concerns like these, which we are asking you to address in your revision, are detailed in each of their Recommendations sections.

3. The Reviewers would like you to add an analysis of the co-expression and segregation dynamics between gene modules for all four ectodermal cell fates (epidermis, placodes, neural crest, neural plate). The Reviewers think this would strengthen the value of your paper.

4. We feel that you should use a different title to reflect the fact that the work is a Tools and Resources manuscript and not a Research Article (which is typically focused on the functional testing of a specific hypothesis). To facilitate the discovery and re-use of your resource, we think your title should instead emphasize the system and methods used and the nature of your dataset.

*Reviewer #1 (Recommendations for the authors):*

To substantiate the claim that cells expressing significant numbers of genes associated with two or more neural plate border lineages retain multi-lineage plasticity (e.g., able to give rise to neural crest and placode derivatives), the authors would at a minimum need to perform lineage tracing. I realize this is not easy in chick embryos, but without such tests, the impressive amount of data and analysis in the paper only makes predictions but does not test them. The authors should revise the text throughout their manuscript to explain these caveats and limitations of their study. For instance, in their Introduction, the authors speak of previous models of border specification and state "They [ the previous studies] therefore do not address how individual cells at the NBP make decisions, nor whether they are multipotent or a population of mixed progenitors that are already biased towards their later identity". The present study does not really address this question either. To give one example, the authors suggest that defined placodal progenitors emerge earlier than neural crest cells. At HH7, do the remaining BLUPs have the potential to give rise to neural crest AND placodes, (to take an extreme), do they all give rise to neural crest despite still expressing some placodal genes? Put another way, how can we tell when the presence of heterogeneity is biologically significant? This should all be addressed in the text.

*Reviewer #2 (Recommendations for the authors):*

I have two suggestions for revision:

First, I think the predictions of the "binary competence model" (sequence of cell-fate decisions: [neural or neural crest]-[epidermal or placodal]) and "neural plate border state model" (sequence of cell fate decisions: neural-[placodal or neural crest]-epidermal) should be highlighted and it should be discussed to what extent the findings of the current study support one or the other model. Most of the data presented by Thiery et al. suggest that cell fate decisions at the NPB occur as predicted by the binary competence model: While placodal cells cluster with epidermal cells, neural crest cells cluster with neural plate cells. Moreover, Figure 4 suggests that the genes of their placodal gene modules are also expressed in epidermal cells but not in the neural plate or neural crest (and – less clearly – neural crest modules are also weakly expressed in the neural plate but not in other ectodermal cells and vice versa). The findings of this paper also support the proposal of the binary competence model that the NPB is an area of indecision that decreases in size over time rather than being defined by its own regulatory state (see in particular Schlosser, 2014). Since neural and epidermal cells are also included in the dataset of this study (except for somite stage 8, where unfortunately no epidermal cells were included), the data could also be used to test the predictions of binary competence and neural plate border state model more directly, by analysing the co-expression and segregation dynamics between gene modules for all four ectodermal cell fates (epidermis, placodes, neural crest, neural plate). In other words: is the increasing segregation of placodal and neural crest gene modules initially accompanied by a co-segregation of epidermal with placodal and of the neural plate with neural crest modules? Such an analysis could help to resolve a long-standing controversy and would strengthen the value of this paper even further.

Second, the issues raised regarding the sequence of specification of the different ectodermal fates should be discussed in light of previous experiments of specification and possible species-specific differences as indicated in my public review.

*Reviewer #3 (Recommendations for the authors):*

The authors describe the neural plate border (NPB) as a transient territory within the neural plate itself, that contains precursors of the neural plate, neural crest, neural placodes, and epidermis. It's not clear to me whether this is the authors' own specific definition for the avian model they are using or if this is a generally accepted definition that applies to both amniotes and anamniotes.

The authors could better define when they think the NPB is truly established for comparisons with other species. I say this because the authors conclude that the NPB should be recognized as an anatomical region of the ectoderm rather than a transcriptional state. But the problem is many embryologists have always regarded the NPB as an anatomical region at the junction between neural and non-neural ectoderm, which is identifiable histologically.

I think a major oversight of the authors' work is not considering it in conjunction with the wealth of lineage-tracing experiments. For example, Bronner Fraser and Fraser (1989) showed through single-cell lineage tracing that a single neural plate cell in a chicken embryo, even at the time the neural tube is closed will give rise to cells that stay in the CNS while also generating neural crest cells. Furthermore, Moury and Jacobson (1990) showed that both neural plate and epidermis which can be morphologically distinguished, dissected from the neural plate border of donor embryos, and transplanted into host embryos, in each case will give rise to neural crest cells. This and other classic data reveal a lot about the potential of neural plate border cells, their ultimate fates and also the developmental time when lineage segregation and cell fate determination occurs. The authors should consider their data and model in light of lineage labeling experiments performed in the Garcia-Castro lab (Basch et al., 2006; Prasad et al., 2020).

In addition, when the dorsal neural folds are ablated, neural crest cells are formed from more ventromedial territories, domains that would appear to violate the authors' segregated NPB domains. Finally, neural crest cells that have already migrated can be transplanted back into the neural plate and give rise to dorsal CNS, or ventral CNS depending on the transplant location, or ultimately migrate again as neural crest cells. I think what all of these studies show, in concert with the rich transcriptional data set the authors have generated, is the considerable heterogeneity of gene expression in NBP cells and their incredible plasticity, which remains for a considerable period of developmental time.

The transcriptional dataset is very rich and will be of value to the field, but I think the authors' overall conclusions are not supported by their data.

The authors make a lot of claims, and naturally much of it has to be quite speculative because its based purely on gene expression or co-expression with genes of known spatiotemporal activity and function. Unfortunately, the bulk of validation is attributed to correlative expression of already well-studied factors which demarcate neural cells from neural crest cells from placode cells from epidermis cells. Despite a lot of data, we learn very little about any new genes and their potential roles in driving the process of specification, lineage selection, and cell fate determination because there's no functional data or test. The authors need to be more forthcoming about the speculative nature of their claims and conclusions.

The authors describe a group of cells as bipotential lineage unstable progenitors (BLUPs), based on the supposition that the cells are transcriptionally unstable. This terminology is a complete misnomer and the authors provide no evidence of transcript instability in these cells. I recommend the authors research the molecular biology literature to appreciate what transcriptional instability really means. What the authors have identified is a limited pool of cells that are bipotential lineage progenitors.

The authors propose a gradient border model for cell fate choice at the neural plate border, but again the choice of word is inaccurate, and because the authors provide no evidence of a gradient in the standard biological sense. This renders the title of the paper incorrect. Perhaps the authors intended to say graded or sequential with reference to the timing of the process. Nonetheless, the schematic summary figure appears to be consistent in principle with the single-cell sequencing data and analysis performed in frog embryos and published in Briggs et al. 2018.

Other concerns/comments:

Inaccurate statements such as that on page 2…"neural crest cells are largely incorporated into the neural folds"… makes no anatomical sense. How can a cell that doesn't exist yet be incorporated into a tissue from which it will delaminate? Perhaps the authors meant neural crest cell progenitors?

Much of the regionalization description was very limited and again a lot of assumptions were based on relatively few genes. For example, the expression of Hoxb1 was noted but apart from expression in distinct cell groups what did the authors glean from its activity? Did the authors consider in their analyses that the anterior is slightly older than the posterior when it comes to interpreting when lineage segregation and cell fate determination occurs?

The authors conclude on page 9 that their analysis predicts that cells move away from cluster boundaries towards definitive cell states, and then on page 11 that neural crest probabilities are only high in late cells reflecting that those defined neural crest cell clusters appear at HH7. But isn't this consistent with what we already know from lineage tracing of neural crest cells in avian embryos? In any case, the sequential combinatorial activation and downregulation of gene expression the authors describe is consistent with key known markers for each of the different lineages but the authors have defined this at a much higher resolution which is a valuable resource for the community.

[Editors' note: further revisions were suggested prior to acceptance, as described below.]

Thank you for resubmitting your work entitled "scRNA-sequencing in chick suggests a probabilistic model for cell fate allocation at the neural plate border" for further consideration by *eLife*. Your revised article has been evaluated by Kathryn Cheah (Senior Editor) and a Reviewing Editor.

The manuscript has been improved but there are some remaining issues that need to be addressed, as outlined below.

*Reviewer #2 (Recommendations for the authors):*

In the revised manuscript, the authors have addressed many of the issues raised by myself and the other two reviewers. In particular, they are now more careful in their terminology in relation to specification (my major second point of criticism). However, in the legend of Figure 6, BLUPs are still called "unstable" rather than "undecided". In line 706, they still write "These findings suggest that these cell states are specified in that order". This should be changed.

The authors also made efforts to address my first major point of criticism regarding the discussion of previous models of cell fate decisions at the neural plate border. They reworded some of the discussion of these models and included some additional analysis of the coexpression of placodal and neural crest modules with different regions of the neural plate. Unfortunately, their data did not include a sufficient number of epidermal cells, so they could not conduct the additional analysis of the sequence of cell fate decisions that I suggested. While these changes have improved the manuscript, I think that the authors still 1) slightly misrepresent the "binary competence model" and 2) fail to discuss an important limitation of their study. As suggested below, these remaining issues can be remedied with relatively minor adjustments to the manuscript.

1) As I already wrote in my comments to the first version, a "neural plate border model" that argues that the neural plate border is a region of indecision is compatible with multiple possible models of how cell fate decisions are made at the neural plate border (and in Schlosser, 2014, I specifically propose a model where the NPB is initially characterized as an area of indecision before it resolves into neural and non-neural competence territories). One of these models addressing the sequence of cell fate decisions is the "binary competence model" that we have previously proposed, in which the sequence of cell-fate decisions is [neural or neural crest]-[epidermal or placodal] (as the authors now mention in their introduction). A second possible model, that we discussed (and rejected) as a possible alternative to the binary competence model is a "neural plate border state model" which suggests a sequence of cell fate decisions: neural-[placodal or neural crest]-epidermal (see Schlosser, 2006, 2014). A third possible alternative is a model in which the four different ectodermal territories may be specified by multiple different pathways of cell fate decision that may not show a clear preference for one or the other hierarchy of cell fate decisions (this is what Roellig et al., 2017 appear to be proposing although it is at odds with their own findings, which show different co-expression frequencies for markers of different ectodermal territories).

Since the "binary competence model" is one of several models addressing the sequence of cell fate decisions at the NPB (all of them compatible with the idea that the NPB may start out as an area of indecision), it is misleading to specifically present the "binary competence model" as the one and only alternative to a "neural plate border" model that suggests "that cells at the border of the neural plate have mixed identity and retain the ability to generate all ectodermal derivatives until after neurulation begins" as the authors write in line 98. Instead, the authors should acknowledge (e.g. in lines 97 and following and in the discussion) that the "binary competence model" has been presented as an alternative to models, which propose a different sequence of cell fate decisions at the NPB, such as neural-[placodal or neural crest]-epidermal.

2) In the discussion (lines 770 and following) the authors should acknowledge that while their data support a model of the neural plate border as an area of indecision, the lack of data on epidermal cells does not allow them to address the question of whether any particular sequence of cell fate decision at the NPB is more likely than another sequence and, thus, does not allow them to decide between the different possible models regarding the probability of particular sequences of cell fate decision at the neural plate border (such as "binary competence" versus "neural plate border state" model).

---

## [Author Response]

Essential revisions:1. The Reviewers had major concerns about the language used to describe your results, your terminology, the interpretations of your findings, and the claims made about the significance of your study. The Reviewers felt that for your claims to be substantiated as written, you would have to perform additional proof-of-concept experiments and confirmatory analyses. Theoretically, these would include new gain- and loss-of-function experiments, lineage tracing analyses, and heterotopic transplantations.However, given that your submission constitutes a Tools and Resources manuscript and not a Research Article, we are not asking you to perform additional experiments. Instead, the Reviewers are asking you to take very seriously all their other their requests (major and minor) for a substantial revision of the abstract and text, and a reanalysis of some of the data as detailed and described in each of their Recommendations sections.

We have addressed the reviewers’ comments and scrutinised our terminology carefully to avoid confusion and included new analysis of our data and revising the text.

2. Specifically, we are asking you to revise your manuscript to address what the Reviewers identify as conceptual problems with the interpretation of your data, unsubstantiated claims about the significance of your results, insufficient discussion of other similar previous studies/lineage tracing experiments, and highly speculative conclusions. For example, the Reviewers take issue with your use of the term BLUPs for transcriptionally unstable cells and your suggestion of a gradient model that they feel lacks molecular evidence. The Reviewers have also asked you to define and use terms like commitment, specification, etc. with more precision. Other concerns like these, which we are asking you to address in your revision, are detailed in each of their Recommendations sections.

We have carefully edited the text to include relevant studies, to clarify the concepts and how our data allows to re-visit these concepts. We agree that we do not know whether BULPs are indeed transcriptionally unstable and have therefore renamed them as ‘border located undecided progenitors’.

3. The Reviewers would like you to add an analysis of the co-expression and segregation dynamics between gene modules for all four ectodermal cell fates (epidermis, placodes, neural crest, neural plate). The Reviewers think this would strengthen the value of your paper.

We thank the reviewers for this suggestion and have added a supplementary figure (Figure 6—figure supplement 3) showing co-expression of other cell fates found adjacent to one another in the embryonic ectoderm (placodal/forebrain and neural crest/mid-hindbrain). This analysis reveals that complementary populations of cells co-express different combinations of definitive lineage gene modules, and in turn the heterogeneity of cell fate decisions taking place along the anterior-posterior axis.

4. We feel that you should use a different title to reflect the fact that the work is a Tools and Resources manuscript and not a Research Article (which is typically focused on the functional testing of a specific hypothesis). To facilitate the discovery and re-use of your resource, we think your title should instead emphasize the system and methods used and the nature of your dataset.

We have re-worded the title accordingly.

Reviewer #1 (Recommendations for the authors):To substantiate the claim that cells expressing significant numbers of genes associated with two or more neural plate border lineages retain multi-lineage plasticity (e.g., able to give rise to neural crest and placode derivatives), the authors would at a minimum need to perform lineage tracing. I realize this is not easy in chick embryos, but without such tests, the impressive amount of data and analysis in the paper only makes predictions but does not test them. The authors should revise the text throughout their manuscript to explain these caveats and limitations of their study. For instance, in their Introduction, the authors speak of previous models of border specification and state "They [ the previous studies] therefore do not address how individual cells at the NBP make decisions, nor whether they are multipotent or a population of mixed progenitors that are already biased towards their later identity". The present study does not really address this question either.

We thank the review for this helpful comment. We agree that we do not address whether cells at the neural plate border are multipotent and have rephrased this section in the introduction to highlight that the focus of our manuscript is on the heterogeneity of cell fate decisions at the neural plate border.

To give one example, the authors suggest that defined placodal progenitors emerge earlier than neural crest cells. At HH7, do the remaining BLUPs have the potential to give rise to neural crest AND placodes, (to take an extreme), do they all give rise to neural crest despite still expressing some placodal genes? Put another way, how can we tell when the presence of heterogeneity is biologically significant? This should all be addressed in the text.

The question of validating the biological significance of transcriptional heterogeneity is of primary interest to us but requires single-cell lineage tracing which is technically challenging and beyond the scope of this study. However, we thank the reviewer for this comment as they are correct that multi-potency can only be validated experimentally. We have revisited the manuscript and made sure that references to cell type emergence and cell state heterogeneity are specifically mentioned in the context of transcriptional state and not competence. We have also made clear in the discussion that testing multipotency requires future single cell lineage tracing.

Reviewer #2 (Recommendations for the authors):I have two suggestions for revision:First, I think the predictions of the "binary competence model" (sequence of cell-fate decisions: [neural or neural crest]-[epidermal or placodal]) and "neural plate border state model" (sequence of cell fate decisions: neural-[placodal or neural crest]-epidermal) should be highlighted and it should be discussed to what extent the findings of the current study support one or the other model. Most of the data presented by Thiery et al. suggest that cell fate decisions at the NPB occur as predicted by the binary competence model: While placodal cells cluster with epidermal cells, neural crest cells cluster with neural plate cells. Moreover, Figure 4 suggests that the genes of their placodal gene modules are also expressed in epidermal cells but not in the neural plate or neural crest (and – less clearly – neural crest modules are also weakly expressed in the neural plate but not in other ectodermal cells and vice versa). The findings of this paper also support the proposal of the binary competence model that the NPB is an area of indecision that decreases in size over time rather than being defined by its own regulatory state (see in particular Schlosser, 2014). Since neural and epidermal cells are also included in the dataset of this study (except for somite stage 8, where unfortunately no epidermal cells were included), the data could also be used to test the predictions of binary competence and neural plate border state model more directly, by analysing the co-expression and segregation dynamics between gene modules for all four ectodermal cell fates (epidermis, placodes, neural crest, neural plate). In other words: is the increasing segregation of placodal and neural crest gene modules initially accompanied by a co-segregation of epidermal with placodal and of the neural plate with neural crest modules? Such an analysis could help to resolve a long-standing controversy and would strengthen the value of this paper even further.

We appreciate these suggestions which have helped us to improve the manuscript. While we were unable to carry out pairwise co-expression analysis of gene modules for all four ectodermal fates due to limited epidermal cell availability at later timepoints, we have conducted additional co-expression analysis of placodal-forebrain and neural crest-mid/hindbrain gene modules. Our findings further support our earlier results indicating a high degree of heterogeneity at the neural plate border. Co-expression of different gene modules differs along that antero-posterior axis reflecting the different kind of fate choices cells have and suggesting a different order depending on cell location.

We acknowledge the usefulness of both the NPB and binary competence models as working hypotheses, but our results suggest that neither model fully accounts for this heterogeneity. We have expanded our Results section to include these additional findings and provided greater clarity in the discussion regarding how our proposed probabilistic model fits within the context of the existing models. We hope that our revised manuscript will better convey our findings and their implications.

Second, the issues raised regarding the sequence of specification of the different ectodermal fates should be discussed in light of previous experiments of specification and possible species-specific differences as indicated in my public review.

As stated above, we have re-worded the text carefully to reflect this comment.

Reviewer #3 (Recommendations for the authors):The authors describe the neural plate border (NPB) as a transient territory within the neural plate itself, that contains precursors of the neural plate, neural crest, neural placodes, and epidermis. It's not clear to me whether this is the authors' own specific definition for the avian model they are using or if this is a generally accepted definition that applies to both amniotes and anamniotes.

To provide clarification, it should be noted that the NPB is not considered a territory within the neural plate, but rather a region located at its edge. This term is commonly used in both amniotes and anamniotes, and it is generally accepted that it contains precursors for all four ectodermal derivatives. The appropriate references are given in the text.

The authors could better define when they think the NPB is truly established for comparisons with other species. I say this because the authors conclude that the NPB should be recognized as an anatomical region of the ectoderm rather than a transcriptional state. But the problem is many embryologists have always regarded the NPB as an anatomical region at the junction between neural and non-neural ectoderm, which is identifiable histologically.

We agree with the reviewer that the NPB is an anatomical location and have clarified this in the text. However, it has generally been assumed that cells within the NPB have the same ‘transcriptional state’. Our data show that this assumption is incorrect and NPB cells should be defined molecularly by the co-expression of competing gene modules. Given the heterogeneity in cells located at the boundary of the neural plate, it is not possible to say when the NPB is truly established. Instead, our detailed co-expression analysis highlights that cells within this region express broad graded co-expression of placodal, neural and neural crest modules, with cell fate allocation taking place gradually between neurulation.

In addition, when the dorsal neural folds are ablated, neural crest cells are formed from more ventromedial territories, domains that would appear to violate the authors' segregated NPB domains. Finally, neural crest cells that have already migrated can be transplanted back into the neural plate and give rise to dorsal CNS, or ventral CNS depending on the transplant location, or ultimately migrate again as neural crest cells. I think what all of these studies show, in concert with the rich transcriptional data set the authors have generated, is the considerable heterogeneity of gene expression in NBP cells and their incredible plasticity, which remains for a considerable period of developmental time.

We appreciate the reviewer’s comments and agree that there is a lot of plasticity even at later developmental stages. Indeed, this is supported by our data and the fact that individual cells continue to co-express competing transcriptional programme at quite late stages. Our data do not suggest that cells are ‘committed’ even when they express gene modules characteristic for a specific fate, and we have carefully screened the text to make this clear.

The transcriptional dataset is very rich and will be of value to the field, but I think the authors' overall conclusions are not supported by their data.The authors make a lot of claims, and naturally much of it has to be quite speculative because its based purely on gene expression or co-expression with genes of known spatiotemporal activity and function. Unfortunately, the bulk of validation is attributed to correlative expression of already well-studied factors which demarcate neural cells from neural crest cells from placode cells from epidermis cells. Despite a lot of data, we learn very little about any new genes and their potential roles in driving the process of specification, lineage selection, and cell fate determination because there's no functional data or test. The authors need to be more forthcoming about the speculative nature of their claims and conclusions.

We agree with the reviewer that we do not provide any validation of our predictions; it is for this reason that we submitted our work as a resource to make our data available to the community.

The authors describe a group of cells as bipotential lineage unstable progenitors (BLUPs), based on the supposition that the cells are transcriptionally unstable. This terminology is a complete misnomer and the authors provide no evidence of transcript instability in these cells. I recommend the authors research the molecular biology literature to appreciate what transcriptional instability really means. What the authors have identified is a limited pool of cells that are bipotential lineage progenitors.

We agree with the reviewer that ‘unstable’ was not the correct term to use. As per suggestion of reviewer 2 we have changed this to ‘undecided’. Our data do not tell us anything about ‘potential’ of progenitors.

The authors propose a gradient border model for cell fate choice at the neural plate border, but again the choice of word is inaccurate, and because the authors provide no evidence of a gradient in the standard biological sense. This renders the title of the paper incorrect. Perhaps the authors intended to say graded or sequential with reference to the timing of the process. Nonetheless, the schematic summary figure appears to be consistent in principle with the single-cell sequencing data and analysis performed in frog embryos and published in Briggs et al. 2018.

We appreciate the reviewer’s comment and agree that ‘gradient’ was misleading. We have now replaced with graded and expanded Figure 6 to highlight the graded co-expression of gene modules associated with alternate fates.

Other concerns/comments:Inaccurate statements such as that on page 2…"neural crest cells are largely incorporated into the neural folds"… makes no anatomical sense. How can a cell that doesn't exist yet be incorporated into a tissue from which it will delaminate? Perhaps the authors meant neural crest cell progenitors?

Thanks for spotting this; we have now modified the statement.

Much of the regionalization description was very limited and again a lot of assumptions were based on relatively few genes. For example, the expression of Hoxb1 was noted but apart from expression in distinct cell groups what did the authors glean from its activity? Did the authors consider in their analyses that the anterior is slightly older than the posterior when it comes to interpreting when lineage segregation and cell fate determination occurs?

The reviewer is correct that we keep the description of anterior-posterior patterning to a minimum; this is not the focus of our paper and we feel that expanding the manuscript to cover this aspect will over-complicate the manuscript and make it even longer.

The authors conclude on page 9 that their analysis predicts that cells move away from cluster boundaries towards definitive cell states, and then on page 11 that neural crest probabilities are only high in late cells reflecting that those defined neural crest cell clusters appear at HH7. But isn't this consistent with what we already know from lineage tracing of neural crest cells in avian embryos? In any case, the sequential combinatorial activation and downregulation of gene expression the authors describe is consistent with key known markers for each of the different lineages but the authors have defined this at a much higher resolution which is a valuable resource for the community.

Our data suggest show that the NCC transcriptional state arises late and therefore the probability is high late. We are not sure how to relate this to lineage tracing in the embryo.

[Editors' note: further revisions were suggested prior to acceptance, as described below.]

The manuscript has been improved but there are some remaining issues that need to be addressed, as outlined below.Reviewer #2 (Recommendations for the authors):In the revised manuscript, the authors have addressed many of the issues raised by myself and the other two reviewers. In particular, they are now more careful in their terminology in relation to specification (my major second point of criticism). However, in the legend of Figure 6, BLUPs are still called "unstable" rather than "undecided". In line 706, they still write "These findings suggest that these cell states are specified in that order". This should be changed.

Apologies, we must have missed this. Now corrected legend and reworded the text to read: “These findings suggest that these cell states are *emerge* in that order.”

The authors also made efforts to address my first major point of criticism regarding the discussion of previous models of cell fate decisions at the neural plate border. They reworded some of the discussion of these models and included some additional analysis of the coexpression of placodal and neural crest modules with different regions of the neural plate. Unfortunately, their data did not include a sufficient number of epidermal cells, so they could not conduct the additional analysis of the sequence of cell fate decisions that I suggested. While these changes have improved the manuscript, I think that the authors still 1) slightly misrepresent the "binary competence model" and 2) fail to discuss an important limitation of their study. As suggested below, these remaining issues can be remedied with relatively minor adjustments to the manuscript.1) As I already wrote in my comments to the first version, a "neural plate border model" that argues that the neural plate border is a region of indecision is compatible with multiple possible models of how cell fate decisions are made at the neural plate border (and in Schlosser, 2014, I specifically propose a model where the NPB is initially characterized as an area of indecision before it resolves into neural and non-neural competence territories). One of these models addressing the sequence of cell fate decisions is the "binary competence model" that we have previously proposed, in which the sequence of cell-fate decisions is [neural or neural crest]-[epidermal or placodal] (as the authors now mention in their introduction). A second possible model, that we discussed (and rejected) as a possible alternative to the binary competence model is a "neural plate border state model" which suggests a sequence of cell fate decisions: neural-[placodal or neural crest]-epidermal (see Schlosser, 2006, 2014). A third possible alternative is a model in which the four different ectodermal territories may be specified by multiple different pathways of cell fate decision that may not show a clear preference for one or the other hierarchy of cell fate decisions (this is what Roellig et al., 2017 appear to be proposing although it is at odds with their own findings, which show different co-expression frequencies for markers of different ectodermal territories).Since the "binary competence model" is one of several models addressing the sequence of cell fate decisions at the NPB (all of them compatible with the idea that the NPB may start out as an area of indecision), it is misleading to specifically present the "binary competence model" as the one and only alternative to a "neural plate border" model that suggests "that cells at the border of the neural plate have mixed identity and retain the ability to generate all ectodermal derivatives until after neurulation begins" as the authors write in line 98. Instead, the authors should acknowledge (e.g. in lines 97 and following and in the discussion) that the "binary competence model" has been presented as an alternative to models, which propose a different sequence of cell fate decisions at the NPB, such as neural-[placodal or neural crest]-epidermal.

We appreciate the comments of the reviewer; we have now added more detail on the recently revised dual competence model (introduction line 87 onwards; discussion line 762 onwards) to include the proposal from Maharana and Schlosser.

However, we respectfully disagree with the suggestion that the NPB model suggests a specific sequence of fate decisions: neural-[placodal or neural crest]-epidermal. While this is stated in the 2006 and 2014 reviews, to the best of our knowledge there is no evidence in the literature to support such a sequence of events (including in the papers cited in the reviews). We are therefore puzzled where this proposal comes from. It is for this reason that we are reluctant to state that the NPB model suggests a specific sequence of fate decisions.

We would like to emphasize that our data do not provide support for either model, but highlight the heterogeneity of cells in the NPB, the fact that cells expressing ‘conflicting’ TF programmes can be found very late even after the time of neural tube closure and that fate decisions themselves are heterogeneous in sequence and place. This is clearly stated in the final section of the discussion.